# Beyond Normal: On the Evaluation of Mutual Information Estimators

**Paweł Czyż**[*1,2]    **Frederic Grabowski**[* 3]
**Julia E. Vogt**[4,5]    **Niko Beerenwinkel**[† 1,5]    **Alexander Marx**[† 2,4]
[1]Department of Biosystems Science and Engineering, ETH Zurich    [2]ETH AI Center, ETH Zurich
[3]Institute of Fundamental Technological Research, Polish Academy of Sciences
[4]Department of Computer Science, ETH Zurich    [5]SIB Swiss Institute of Bioinformatics

## Abstract

Mutual information is a general statistical dependency measure which has found applications in representation learning, causality, domain generalization and computational biology. However, mutual information estimators are typically evaluated on simple families of probability distributions, namely multivariate normal distribution and selected distributions with one-dimensional random variables. In this paper, we show how to construct a diverse family of distributions with known ground-truth mutual information and propose a language-independent benchmarking platform for mutual information estimators. We discuss the general applicability and limitations of classical and neural estimators in settings involving high dimensions, sparse interactions, long-tailed distributions, and high mutual information. Finally, we provide guidelines for practitioners on how to select appropriate estimator adapted to the difficulty of problem considered and issues one needs to consider when applying an estimator to a new data set.

## 1   Introduction

Estimating the strength of a non-linear dependence between two continuous random variables (r.v.) lies at the core of machine learning. Mutual information (MI) lends itself naturally to this task, due to its desirable properties, such as invariance to homeomorphisms and the data processing inequality. It finds applications in domain generalization [Li et al., 2022, Ragonesi et al., 2021], representation learning [Belghazi et al., 2018, Oord et al., 2018], causality [Solo, 2008, Kurutach et al., 2018], physics [Keys et al., 2015], systems biology [Selimkhanov et al., 2014, Grabowski et al., 2019, Uda, 2020], and epidemiology [Young et al., 2023].

Over the last decades, the estimation of MI has been extensively studied, and estimators have been developed, ranging from classical approaches based on histogram density [Pizer et al., 1987], kernel density estimation [Moon et al., 1995] to $k$-nearest neighbor [Kozachenko and Leonenko, 1987, Kraskov et al., 2004] and neural estimators [Belghazi et al., 2018, Oord et al., 2018, Song and Ermon, 2020]. However, despite the progress that has been achieved in this area, not much attention has been focused on systematically benchmarking these approaches.

Typically, new estimators are evaluated assuming multivariate normal distributions for which MI is analytically tractable [Darbellay and Vajda, 1999, Kraskov et al., 2004, Suzuki, 2016]. Sometimes, simple transformations are applied to the data [Khan et al., 2007, Gao et al., 2015], moderately high-dimensional settings are considered [Lord et al., 2018, Lu and Peltonen, 2020], or strong dependencies are evaluated [Gao et al., 2015]. Beyond that, Song and Ermon [2020] study the

---

*Equal contribution [†]Joint supervision

37th Conference on Neural Information Processing Systems (NeurIPS 2023).

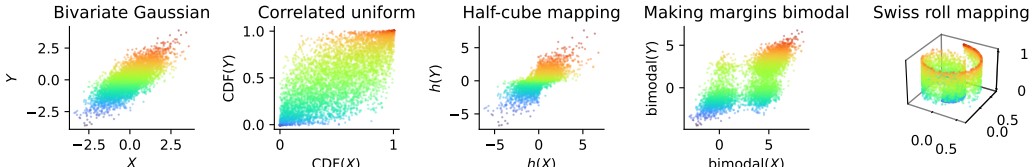

Figure 1: Visualisations of selected proposed distributions. Two correlated Gaussian variables $X$ and $Y$ (1) can be transformed via the Gaussian CDF into correlated uniform variables (2), into a long-tailed distribution via the "half-cube" mapping $t \mapsto t\sqrt{|t|}$ (3), into a multi-modal distribution (4) or embedded in the three-dimensional space via the composition of the Swiss roll mapping and Gaussian CDF (5). Color on the plot corresponds to the original $Y$ variable.

self-consistency (additivity under independence and the data processing inequality) of neural MI estimators in the context of representation learning from image data.

**Our Contributions** In this work we show a method of developing expressive distributions with known ground-truth mutual information (Sec. 2), propose forty benchmark tasks and systematically study the properties of commonly used estimators, including representatives based on kernel or histogram density estimation, $k$NN estimation, and neural network-based estimators (Sec. 3). We address selected difficulties one can encounter when estimating mutual information (Sec. 4), such as sparsity of interactions, long-tailed distributions, invariance, and high mutual information. Finally, we provide recommendations for practitioners on how to choose a suitable estimator for particular problems (Sec. 6). Our benchmark is designed so that it is simple to add new tasks and estimators. It also allows for cross-language comparisons — in our work we compare estimators implemented in Python, R and Julia. All of our experiments are fully reproducible by running Snakemake workflows [Mölder et al., 2021]. Accompanying code is available at `http://github.com/cbg-ethz/bmi`.

Overall, our *key findings* from the benchmark can be summarized as follows:

- Testing on multivariate normal distributions gives a biased and overly optimistic view of estimator performance. In this setting, canonical correlation analysis (CCA), a model-based approach, emerges as the most effective method — even if the model is slightly misspecified.
- Compared to classical estimators, neural estimators excel in high-dimensional settings, capturing sparse interactions across multiple dimensions.
- The popular KSG estimator [Kraskov et al., 2004] is accurate in low- and moderate-dimensional settings, but its performance suffers on problems involving high-dimensions or sparse interactions.
- Although MI is invariant to a wide range of transformations (Theorem 2.1), numerical estimates, even with large sample sizes, are not. Thus, MI may not be suitable when metric invariance to specific transformation families is required.
- Multivariate Student distributions pose an important and hard challenge to many mutual information estimators. This effect is partially, but not fully, attributed to their long tails.

## 2 Mutual Information: Estimation and Invariance

We start by recalling the definition of MI, then discuss the problem of estimating it, and introduce the estimators used in the benchmark.

Consider two r.v. $X$ and $Y$ with domains $\mathcal{X}$ respectively $\mathcal{Y}$, joint probability measure $P_{XY}$ and marginal probability measures $P_X$ and $P_Y$, respectively. If $P_{XY}$ is absolutely continuous with respect to $P_X \otimes P_Y$ (e.g., $X$ and $Y$ are finite r.v.), MI is equal to the following Kullback–Leibler divergence[2]:

$$\mathbf{I}(X;Y) = \mathbf{D}_{\mathrm{KL}}\left(P_{XY} \parallel P_X \otimes P_Y\right) = \int \log f \, \mathrm{d}P_{XY},$$

where $f = \mathrm{d}P_{XY}/\mathrm{d}(P_X \otimes P_Y)$ is the Radon–Nikodym derivative. If the absolute continuity does not hold, then $\mathbf{I}(X;Y) = +\infty$ [Pinsker and Feinstein, 1964, Theorem 2.1.2]. If $\mathcal{X}$ and $\mathcal{Y}$ are Euclidean spaces and measures $P_{XY}$, $P_X$ and $P_Y$ have probability density functions (PDFs) with respect to the Lebesgue measure, then the Kullback–Leibler divergence can be written in terms of the PDFs.

---

[2]We use the natural logarithm, meaning that the mutual information is measured in *nats*.

However, we will later consider distributions which do not have PDFs with respect to the Lebesgue measure and the general Radon–Nikodym derivative must be used (see "Swiss roll" in Fig. 1).

In almost all applications, $P_{XY}$ and $P_X \otimes P_Y$ are not known and one needs to estimate MI from a finite sample from $P_{XY}$, i.e., a realization $\big((x_1, y_1), \ldots, (x_N, y_N)\big) \in (\mathcal{X} \times \mathcal{Y})^N$ of $N$ i.i.d. r.v. $(X_1, Y_1), \ldots, (X_N, Y_N)$ distributed according to the joint distribution $P_{XY}$. As a MI estimator we will denote a family of measurable mappings, indexed by the sample size $N$ and spaces $\mathcal{X}$ and $\mathcal{Y}$ (in most applications, Euclidean spaces of various dimensionalities): $e_N^{\mathcal{X}\mathcal{Y}} \colon (\mathcal{X} \times \mathcal{Y})^N \to \mathbb{R}$. Composing this mapping with the r.v. representing the whole data set we obtain a real-valued r.v. $E_N^{XY}$. For a given $P_{XY}$ we can consider the distribution of $E_N^{XY}$. It is often summarized by its first two moments, resulting in the bias and the variance of the estimator. Understanding the bias of an estimator however requires the knowledge of ground-truth MI, so the estimators are typically tested on the families of jointly multivariate normal or uniform distributions, with varying $N$ and ground-truth $\mathbf{I}(X; Y)$ [Khan et al., 2007, Lord et al., 2018, Holmes and Nemenman, 2019, Song and Ermon, 2020].

**Constructing Expressive Distributions and Benchmarking** Besides considering different types of distributions such as (multivariate) normal, uniform and Student, our benchmark is based on the invariance of MI to the family of chosen mappings. Generally MI is not invariant to arbitrary transformations, as due to the data processing inequality $\mathbf{I}(f(X); g(Y)) \leq \mathbf{I}(X; Y)$. However, under some circumstances MI is preserved, which enables us to create more expressive distributions by transforming the variables.

**Theorem 2.1.** *Let $\mathcal{X}$, $\mathcal{X}'$, $\mathcal{Y}$ and $\mathcal{Y}'$ be standard Borel spaces (e.g., smooth manifolds with their Borel $\sigma$-algebras) and $f \colon \mathcal{X} \to \mathcal{X}'$ and $g \colon \mathcal{Y} \to \mathcal{Y}'$ be continuous injective mappings. Then, for every $\mathcal{X}$-valued r.v. $X$ and $\mathcal{Y}$-valued r.v. $Y$ it holds that $\mathbf{I}(X; Y) = \mathbf{I}(f(X); g(Y))$.*

This theorem has been studied before by Kraskov et al. [2004, Appendix], who consider diffeomorphisms and by Polyanskiy and Wu [2022, Th. 3.7], who assume measurable injective functions with measurable left inverses. For completeness we provide a proof which covers continuous injective mappings in Appendix A. In particular, each of $f$ and $g$ can be a homeomorphism, a diffeomorphism, or a topological embedding. Embeddings are allowed to increase the dimensionality of the space, so even if $X$ and $Y$ have probability density functions, $f(X)$ and $g(Y)$ do not need to have them. Neither of these deformations change MI. Thus, using Theorem 2.1, we can create more expressive distributions $P_{f(X)g(Y)}$ by sampling from $P_{XY}$ and transforming the samples to obtain a data set $\big((f(x_1), g(y_1)), \ldots, (f(x_N), g(y_N))\big) \in (\mathcal{X}' \times \mathcal{Y}')^N$.

While offering a tool to construct expressive distributions, another valuable perspective to consider with this problem is estimation invariance: although MI is invariant to proposed changes, the estimate may not be. More precisely, applying an estimator to the transformed data set results in the induced r.v. $E_N^{f(X)g(Y)}$ and its distribution. If the estimator were truly invariant, one would expect that $\mathbb{E}\left[E_N^{f(X)g(Y)}\right]$ should equal $\mathbb{E}\left[E_N^{XY}\right]$. This invariance is rarely questioned and often implicitly assumed as given (see e.g. Tschannen et al. [2020] or Murphy [2023, Sec. 32.2.2.3]), but as we will see in Sec. 4.3, finite-sample estimates are not generally invariant.

# 3 Proposed Benchmark

In this section, we outline forty tasks included in the benchmark that cover a wide family of distributions, including varying tail behaviour, sparsity of interactions, multiple modes in PDF, and transformations that break colinearity. As our base distributions, we selected multivariate normal and Student distributions, which were transformed with continuous injective mappings.

We decided to focus on the following phenomena:

1. **Dimensionality.** High-dimensional data are collected in machine learning and natural sciences [Bühlmann and van de Geer, 2011, Ch. 1]. We therefore change the dimensions of the $\mathcal{X}$ and $\mathcal{Y}$ spaces between 1 and 25.

2. **Sparsity.** While the data might be high-dimensional, the effective dimension may be much smaller, due to correlations between different dimensions or sparsity [Lucas et al., 2006]. We therefore include distributions in which some of the dimensions represent random noise,

which does not contribute to the mutual information, and perform an additional study in Section 4.1.

3. **Varying MI.** Estimating high MI is known to be difficult [McAllester and Stratos, 2020]. However, we can often bound it in advance — if there are 4000 image classes, MI between image class and representation is at most $\log 4000 \approx 8.3$ nats. In this section, we focus on problems with MI varying up to 2 nats. We additionally consider distributions with higher MI in Section 4.4.

4. **Long tails.** As Taleb [2020] and Zhang et al. [2021] argue, many real-world distributions have long tails. To model different tails we consider multivariate Student distributions as well as transformations lengthening the tails. We conduct an additional study in Section 4.2.

5. **Robustness to diffeomorphisms.** As stated in Theorem 2.1, mutual information is invariant to reparametrizations of random variables by diffeomorphisms. We however argue that when only finite samples are available, invariance in mutual information estimates may not be achieved. To test this hypothesis we include distributions obtained by using a diverse set of diffeomorphisms and continuous injective mappings. We further study the robustness to reparametrizations in Section 4.3.

While we provide a concise description here, precise experimental details can be found in Appendix D and we visualise selected distributions in Appendix F. We first describe tasks employing one-dimensional variables.

**Bivariate Normal** For multivariate normal variables MI depends only on the correlation matrix. We will therefore consider a centered bivariate normal distribution $P_{XY}$ with $\mathrm{Cor}(X, Y) = \rho$ and $\mathbf{I}(X; Y) = -0.5 \log \left(1 - \rho^2\right)$. We chose $\rho = 0.75$.

**Uniform Margins** As a first transformation, we apply the Gaussian CDF $F$ to Gaussian r.v. $X$ and $Y$ to obtain r.v. $X' = F(X)$ and $Y' = F(Y)$. It is a standard result in copula theory (see Lemma D.2 or e.g., Nelsen [2007] for an introduction to copulas) that $F(X) \sim \mathrm{Uniform}(0, 1)$ resp. $F(Y) \sim \mathrm{Uniform}(0, 1)$. The joint distribution $P_{X'Y'}$, however, is not uniform. Mutual information is preserved, $\mathbf{I}(X'; Y') = \mathbf{I}(X; Y)$. For an illustration, see Fig. 1. A distribution P transformed with Gaussian CDF $F$ will be denoted as `Normal CDF @ P`.

**Half-Cube Map** To lengthen the tails we applied the half-cube homeomorphism $h(x) = |x|^{3/2} \operatorname{sign} x$ to Gaussian variables $X$ and $Y$. We visualize an example in Fig. 1 and denote a transformed distribution as `Half-cube @ P`.

**Asinh Mapping** To shorten the tails we applied inverse hyperbolic sine function $\operatorname{asinh} x = \log \left(x + \sqrt{1 + x^2}\right)$. A distribution transformed with this mapping will be denoted as `Asinh @ P`.

**Wiggly Mapping** To model non-uniform lengthscales, we applied a mapping

$$w(x) = x + \sum_i a_i \sin(\omega_i x + \varphi_i), \qquad \sum_i |a_i \omega_i| < 1.$$

Due to the inequality constraint this mapping is injective and preserves MI. The parameter values can be found in Appendix D. We denote the transformed distribution as `Wiggly @ P`.

**Bimodal Variables** Applying the inverse CDF of a two-component Gaussian mixture model to correlated variables $X$ and $Y$ with uniform margins we obtained a joint probability distribution $P_{X'Y'}$ with four modes (presented in Fig. 1). We call this distribution `Bimodal` and provide technical details in Appendix D.

**Additive Noise** Consider independent r.v. $X \sim \mathrm{Uniform}(0, 1)$ and $N \sim \mathrm{Uniform}(-\varepsilon, \varepsilon)$, where $\varepsilon$ is the noise level. For $Y = X + N$, it is possible to derive $\mathbf{I}(X; Y)$ analytically (see Appendix D for the formula and the parameter values) and we will call this distribution `Uniform (additive noise=`$\varepsilon$`)`.

**Swiss Roll Embedding** As the last example in this section, we consider a mixed case in which a one-dimensional r.v. $X \sim \mathrm{Uniform}(0, 1)$ is smoothly embedded into two dimensions via the Swiss roll mapping, a popular embedding used to test dimensionality reduction algorithms (see Fig. 1 for visualisation and Appendix D for the formula). Note that the obtained distribution does not have a PDF with respect to the Lebesgue measure on $\mathbb{R}^3$.

Next, we describe the tasks based on multivariate distributions.

**Multivariate Normal**  We sampled $(X, Y) = (X_1, \ldots, X_m, Y_1, \ldots, Y_n)$ from the multivariate normal distribution. We model two correlation structures: "dense" interactions with all off-diagonal correlations set to 0.5 and "2-pair" interactions where $\mathrm{Cor}(X_1, Y_1) = \mathrm{Cor}(X_2, Y_2) = 0.8$ and there is no correlation between any other (distinct) variables (`Multinormal (2-pair)`). For a latent-variable interpretation of these covariance matrices we refer to Appendix D.2.

**Multivariate Student**  To see how well the estimators can capture MI contained in the tails of the distribution, we decided to use multivariate Student distributions (see Appendix D) with dispersion matrix[3] set to the identity $I_{m+n}$ and $\nu$ degrees of freedom. Contrary to the multivariate normal case, in which the identity matrix used as the covariance would lead to zero MI, the variables $X$ and $Y$ can still interact through the tails.

**Transformed Multivariate Distributions**  As the last set of benchmark tasks we decided to transform the multivariate normal and multivariate Student distributions. We apply mappings described above to each axis separately. For example, applying normal CDF to the multivariate normal distribution will map it into a distribution over the cube $(0, 1)^{m+n}$ with uniform marginals. To mix different axes we used a spiral diffeomorphism (denoted as `Spiral @ P`); we defer the exact construction to Appendix D but we visualise this transformation in Fig. 5.

**Estimators**  In our benchmark, we include a diverse set of estimators. Following recent interest, we include four neural estimators, i.e., Donsker–Varadhan (D-V) and MINE [Belghazi et al., 2018], InfoNCE [Oord et al., 2018], and the NWJ estimator [Nguyen et al., 2007, Nowozin et al., 2016, Poole et al., 2019]. As a representative for model-based estimators, we implement an estimator based on canonical correlation analysis (CCA) [Murphy, 2023, Ch. 28], which assumes that the joint distribution is multivariate normal. For more classical estimators, we include the Kraskov, Stögbauer and Grassberger (KSG) estimator [Kraskov et al., 2004] as the most well-known $k$-nearest neighbour-based estimator, a recently proposed kernel-based estimator (LNN) [Gao et al., 2017b], as well as an estimator based on histograms and transfer entropy (both implemented in Julia's `TransferEntropy` library). A detailed overview of the different classes of estimators is provided in Appendix C. Further, we describe the the hyperparameter selection in Appendix E.

**Preprocessing**  As a data preprocessing step, we standardize each dimension using the mean and standard deviation calculated from the sample. Other preprocessing stategies are also possible, but we did not observe a consistent improvement of one preprocessing strategy over the others (see Appendix E).

### 3.1  Benchmark Results

We show the results for $N = 10\,000$ data points in Fig. 2. Neural estimators and KSG perform better than alternative estimators on most problems, with KSG having often low sample requirements (see Appendix E). The simple model-based estimator CCA obtained excellent performance at low sample sizes (see Appendix E), even for slightly transformed multivariate normal distributions. Finally, LNN and the two Julia estimators (histogram and transfer entropy), work well in low dimension but are not viable in medium- and high-dimensional problems ($\dim X + \dim Y \geq 4$).

Overall, we observed that for most $1{\times}1$-dimensional and multivariate normal problems MI can be reliably estimated. Difficulties arise for sparse interactions, atypical (Student) distributions, and for significantly transformed distributions. This suggests that the typical evaluations of MI estimators are overly optimistic, focusing on relatively simple problems.

KSG is able to accurately estimate MI in multivariate normal problems, however performance drops severely on tasks with sparse interactions (2-pair tasks). For $5{\times}5$ dimensions the estimate is about 70% of the true MI, and for $25{\times}25$ dimensions it drops to 10%, which is consistent with previous observations [Marx and Fischer, 2022]. Meanwhile, performance of neural estimators is stable. We study the effect of sparsity in more detail in Sec. 4.

Student distributions have proven challenging to all estimators. This is partially due to the fact that long tails can limit estimation quality (see Sec. 4), and indeed, applying a tail-shortening $\mathrm{asinh}$ transformation improves performance. However, performance still remains far below multivariate normal distributions with similar dimensionality and information content, showing that long tails are

---

[3]The dispersion matrix of multivariate Student distribution is different from its covariance, which does not exist for $\nu \leq 2$.

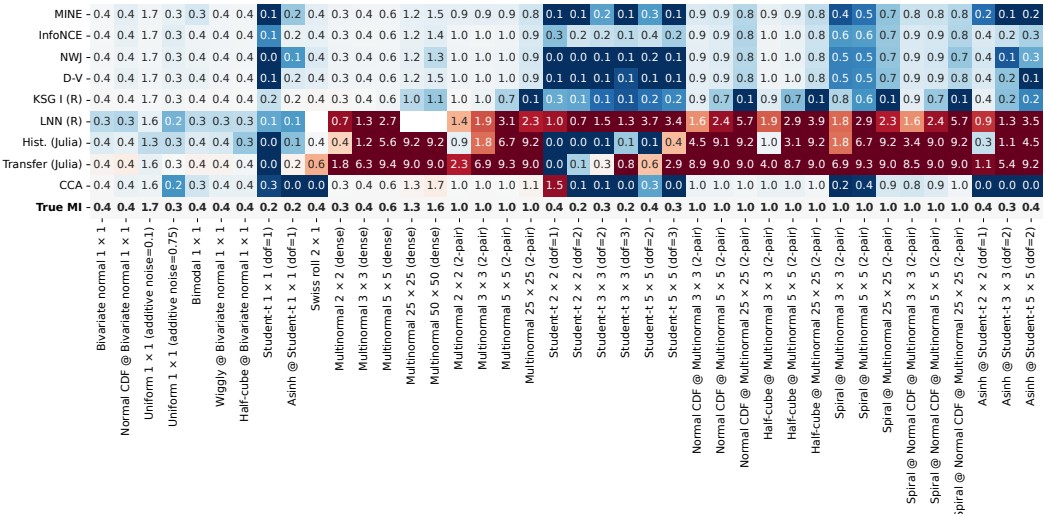

Figure 2: Mean MI estimates of nine estimators over $n = 10$ samples with $N = 10\,000$ points each against the ground-truth value on all benchmark tasks grouped by category. Color indicates relative negative bias (blue) and positive bias (red). Blank entries indicate that an estimator experienced numerical instabilities.

only part of the difficulty. For neural estimators, the issue could be explained by the fact that the critic is not able to fully learn the pointwise mutual information (see Appendix E).

Interestingly, the performance of CCA, which assumes a linear Gaussian model, is often favorable compared to other approaches and has very low sample complexity (see Appendix E). This is surprising, since in the case of transformed distributions the model assumptions are not met. The approach fails for Student distributions (for which the correlation matrix either does not exist or is the identity matrix), the Swiss roll embedding and multivariate normal distributions transformed using the spiral diffeomorphism. Since the spiral diffeomorphism proved to be the most challenging transformation, we study it more carefully in Sec. 4.

## 4 Selected Challenges

In this section we investigate distributions which proved to be particularly challenging for the tested estimators in our benchmark: sparse interactions, long tailed distributions and invariance to diffeomorphisms. Additionally we investigate how the estimators perform for high ground truth MI.

### 4.1 Sparsity of Interactions

In the main benchmark (Fig. 2) we observed that estimation of MI with "2-pair" type of interactions is considerably harder for the KSG estimator than the "dense" interactions. On the other hand, neural estimators were able to pick the relevant interacting dimensions and provide better estimates of MI in these sparse cases.

We decided to interpolate between the "dense" and "2-pair" interaction structures using a two-stage procedure with real parameters $\alpha$ and $\lambda$ and the number of strongly interacting components $K$. While the precise description is in Appendix D.2, we can interpret $\alpha$ as controlling the baseline strength of interaction between every pair of variables in $\{X_1, \ldots, X_{10}, Y_1, \ldots Y_{10}\}$ and $\lambda$ as the additional strength of interaction between pairs of variables $(X_1, Y_1), \ldots, (X_K, Y_K)$. Dense interaction structure has $\lambda = 0$ and 2-pair interaction structure has $\alpha = 0$ and $K = 2$. First, we set $K = 10$ and $\lambda \approx 0$ and decrease $\alpha$ raising $\lambda$ at the same time to maintain constant MI of 1 nat. When $\alpha = 0$, whole information is contained in the pairwise interactions $(X_1, Y_1), \ldots, (X_{10}, Y_{10})$. We then decrease $K$ and increase $\lambda$, still maintaining constant MI.

In Fig. 3 we observe that the performance of all estimators considered (apart from CCA, a model-based approach suitable for multivariate normal distributions) degrades when the interactions between

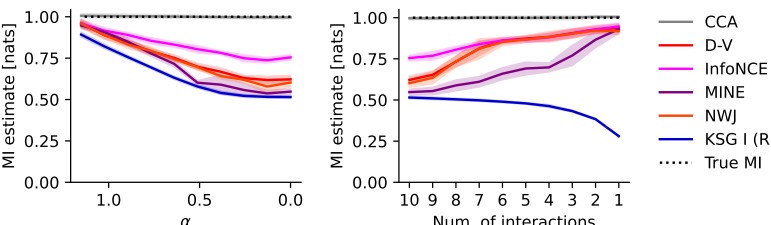

Figure 3: Two-stage interpolation between dense and sparse matrices. From left to right the sparsity of interactions increases. Shaded regions represent the sample standard deviation.

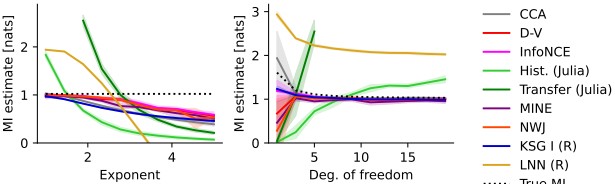

Figure 4: MI estimates as a function of the information contained in the tail of the distribution. Left: lenghtening the tails via $x \mapsto |x|^k \operatorname{sign} x$ mapping with changing $k$. Right: increasing the degrees of freedom in multivariate Student distribution shortens the tails. Shaded regions represent the sample standard deviation.

every pair of variables become sparser (lower $\alpha$). In particular, even neural estimators are not able to model the information in ten interacting pairs of variables. However, when we decrease the number of interacting pairs of variables, the performance of the neighborhood-based KSG estimator is qualitatively different from the neural estimators: performance of KSG steadily deteriorates, while neural estimators can find (some of the) relevant pairs of variables.

This motivates us to conclude that in the settings where considered variables are high-dimensional and the interaction structure is sparse, neural estimators can offer an advantage over neighborhood-based approaches.

## 4.2 Long-Tailed Distributions

In Fig. 4 we investigate two different ways to lengthen the tails. First, we lengthen the tails of a multivariate normal distribution using the mapping $x \mapsto |x|^k \operatorname{sign} x$ (with $k \geq 1$) applied to each dimension separately. In this case, we see that the performance of CCA, KSG, and neural estimators is near-perfect for $k$ close to 1 (when the distribution $P_{XY}$ is close to multivariate normal), but the performance degrades as the exponent increases. Second, we study multivariate Student distributions varying the degrees of freedom: the larger the degrees of freedom, the lower the information content in the tails of the distribution. Again, we see that that the performance of CCA, KSG, and neural estimators is near-perfect for high degrees of freedom for which the distribution is approximately normal. For low degrees of freedom, these estimators significantly underestimate MI, with the exception of CCA which gives estimates with high variance, likely due to the correlation matrix being hard to estimate.

Overall, we see that long tails complicate the process of estimating MI. The Student distribution is particularly interesting, since even after the tails are removed (with $\operatorname{asinh}$ or preprocessing strategies described in Appendix E) the task remains challenging. We suspect that the MI might be contained in regions which are rarely sampled, making the MI hard to estimate. When neural estimators are used, pointwise mutual information learned by the critic does not match the ground truth (see Appendix E).

Hence, we believe that accurate estimation of MI in long-tailed distributions is an open problem. Practitioners expecting long-tailed distributions should approach this task with caution.

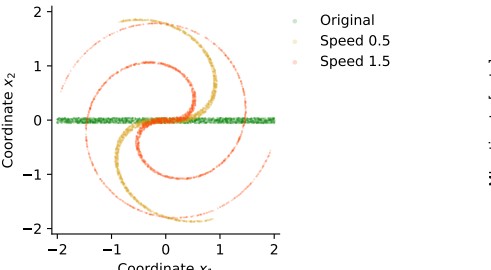 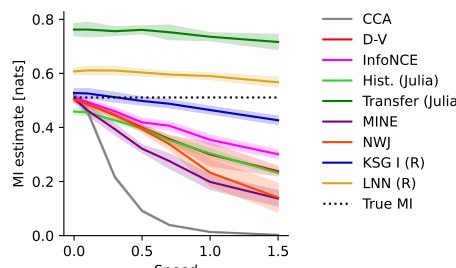

Figure 5: Left: Spiral diffeomorphism with different speed parameter $v$. Right: Performance of considered estimators for increasing speed.

### 4.3 Invariance to Diffeomorphisms

The benchmark results (Fig. 2) as well as the experiments with function $x \mapsto |x|^k \operatorname{sign} x$ lengthening tails (Fig. 4) suggest that even if ground-truth MI is preserved by a transformation (see Theorem 2.1), the finite-sample estimates may not be invariant. This can be demonstrated using the spiral diffeomorphism.

We consider isotropic multivariate normal variables $X = (X_1, X_2) \sim \mathcal{N}(0, I_2)$ and $Y \sim \mathcal{N}(0, 1)$ with $\operatorname{Cor}(X_1, Y) = \rho$. Then, we apply a spiral diffeomorphism $s_v \colon \mathbb{R}^2 \to \mathbb{R}^2$ to $X$, where

$$s_v(x) = \begin{pmatrix} \cos v\|x\|^2 & -\sin v\|x\|^2 \\ \sin v\|x\|^2 & \cos v\|x\|^2 \end{pmatrix}$$

is the rotation around the origin by the angle $v\|x\|^2$ (see Fig. 5; for a theoretical study of spiral diffeomorphisms see Appendix B). In Fig. 5, we present the estimates for $N = 10\,000$, $\rho = 0.8$ and varying speed $v$.

In general, the performance of the considered estimators deteriorates with stronger deformations. This is especially visible for the CCA estimator, as MI cannot be reliably estimated in terms of correlation.

### 4.4 High Mutual Information

Gao et al. [2015] proves that estimation of MI between strongly interacting variables requires large sample sizes. Poole et al. [2019], Song and Ermon [2020] investigate estimation of high MI using multivariate normal distributions in the setting when the ground-truth MI is changing over the training time. We decided to use our family of expressive distributions to see the practical limitations of estimating MI. We chose three one-parameter distribution families: the $3 \times 3$ multivariate normal distribution with correlation $\rho = \operatorname{Cor}(X_1, Y_1) = \operatorname{Cor}(X_2, Y_2)$, and its transformations by the half-cube mapping and the spiral diffeomorphism with $v = 1/3$. We changed the parameter to match the desired MI.

In Fig. 6 we see that for low enough MI neural estimators and KSG reliably estimate MI for all considered distributions, but the moment at which the estimation becomes inaccurate depends on the distribution. For example, for Gaussian distributions, neural estimators (and the simple CCA baseline) are accurate even up to 5 nats, while after the application of the spiral diffeomorphism, the estimates become inaccurate already at 2 nats.

We see that in general neural estimators perform well on distributions with high MI. The performance of CCA on the multivariate normal distribution suggests that well-specified model-based estimators may require lower number of samples to estimate high mutual information than model-free alternatives (cf. Gao et al. [2015]).

## 5 Related Work

**Existing Benchmarks**  Khan et al. [2007] used one-dimensional variables and an additive noise assumption to evaluate the estimators based on histograms, nearest neighbors, and kernels. However,

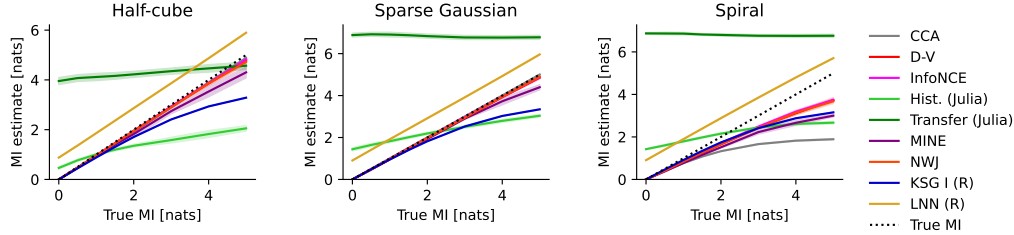

Figure 6: We select three one-parameter families of distributions varying the MI value ($x$-axis and dashed line). We use $N = 10\,000$ data points and $n = 5$ replicates of each experiment.

Table 1: Comparison between benchmarks.

| | Khan et al. [2007] | Poole et al. [2019] | Song and Ermon [2020] | Proposed Benchmark |
|---|---|---|---|---|
| Gaussian Distrib. | ✓ | ✓ | ✓ | ✓ |
| Long-Tailed Distrib. | (✓) | (✓) | ✗ | ✓ |
| Sparse/Dense Inter. | ✗ | ✗ | ✗ | ✓ |
| High MI | ✗ | (✓) | (✓) | ✓ |
| Invariance to Homeom. | ✗ | ✗ | ✗ | ✓ |
| Self-Consistency | ✗ | ✗ | ✓ | ✗ |
| Language Indep. | ✗ | ✗ | ✗ | ✓ |
| Code Available | ✗ | ✗ | ✓ | ✓ |

the original code and samples are not available. Song and Ermon [2020] proposed to evaluate neural estimators based on variational lower bounds [Oord et al., 2018, Belghazi et al., 2018] on image data. However, besides the multivariate Gaussian which they also evaluate, the ground-truth MI is not available. Therefore, they proposed self-consistency checks, which include $\mathbf{I}(X;Y){=}0$ for independent $X$ and $Y$, and preservation of the data processing inequality. Poole et al. [2019] study neural estimators in settings employing Gaussian variables, affine transformation followed by axis-wise cubic transformation and investigate the high MI cases. In our benchmark, we focus on distributions with known MI and information-preserving continuous injective mappings such as homeomorphisms, evaluate long tail distributions, and compare representative estimators from different classes and platforms. We summarize the differences between the benchmarks in Table 1.[4]

**Causal Representation Learning and Nonlinear ICA** Recent work on disentanglement and non-linear independent component analysis (ICA) aims at identifying causal sources up to an ambiguous transformation [Xi and Bloem-Reddy, 2022]. Examples include orthogonal transformations [Zimmermann et al., 2021], permutations and diffeomorphisms applied along specific axes [Khemakhem et al., 2020], or the whole diffeomorphism group [Von Kügelgen et al., 2021, Daunhawer et al., 2023]. To check whether the learned representations indeed capture underlying causal variables, it is important to use estimators which are invariant to the ambiguous transformations specific to the model. Our results suggest that such requirements are not met by MI estimators and practitioners should carefully choose the used metrics, so they are not only theoretically invariant, but also can be robustly estimated in practice.

## 6 Conclusions and Limitations

Venturing beyond the typical evaluation of mutual information estimators on multivariate normal distributions, we provided evidence that the evaluation on Gaussian distributions results in overly optimistic and biased results for existing estimators. In particular, a simple model-based approach as

---

[4]Khan et al. [2007] consider two one-dimensional r.v. related by additive noise, in which one of the r.v. is parametrized with a cubic function. This implicitly lengthens the tails, so we marked this with (✓); similarly, Poole et al. [2019] consider cubic transformation of a multivariate normal variable. The experimental setup of estimation of MI considered by Poole et al. [2019] and Song and Ermon [2020] requires changing ground truth MI over training time of a neural estimator. This procedure differs from the usual approaches to estimating MI, so we marked it with (✓).

CCA is already sufficient if the distribution is (close to) multivariate normal, highlighting the effect of using available prior information about the problem to design efficient estimators. To provide a basis for more thorough evaluation, we designed a benchmark, compatible with any programming language, with forty general tasks focusing on selected problems, as estimation with sparse interactions, the effects of long tails, estimation in problems with high MI, and (only theoretical) invariance to diffeomorphisms.

Our results suggest that in low- and medium-dimensional problems the KSG estimator remains a solid baseline, requiring a lower number of samples than the alternatives and no parameter tuning. In high-dimensional settings where the interactions are sparse, neural estimators offer an advantage over classical estimators being able to more consistently pick up the signal hidden in the high-dimensional space. We further provide evidence that although MI is invariant to continuous injective transformations of variables, in practice the estimates can differ substantially. Appropriate data preprocessing may mitigate some of these issues, but no strategy clearly outperforms the others (see Appendix E).

We hope that the proposed benchmark encourages the development of new estimators, which address the challenges that we highlight in this paper. It can be used to assess invariance to homeomorphisms, understand an estimator's behavior in the situations involving distributions of different complexity, low-data behaviour, high mutual information or to diagnose problems with implementations. We believe that some parts of the benchmark could also be used to assess whether other estimators of other statistical quantities, such as kernel dependence tests, transfer entropy or representation similarity measures, can be reliably estimated on synthetic problems and if they are robust to selected classes of transformations.

**Limitations and Open Problems**  Although the proposed benchmark covers a wide family of distributions, its main limitation is the possibility to only transform the distribution $P_{XY}$ via transformations $f \times g$ to $P_{f(X)g(Y)}$ starting from multivariate normal and Student distributions. In particular, not every possible joint distribution is of this form and extending the family of distributions which have known MI and allow efficient sampling is the natural direction of continuation. We are, however, confident that due to the code design it will be easy to extend the benchmark with the new distributions (and their transformations) when they appear.

Estimation of information in multivariate Student distributions remains a challenge for most estimators. Although we provide the evidence that distributions with longer tails pose a harder challenge for considered estimators, shortening the tails with asinh transform did not alleviate all the issues. In neural estimators this can be partly explained by the fact that the critic may not learn to approximate pointwise mutual information (up to an additive constant) properly, as demonstrated in Appendix E, but the question of what makes the neighborhood-based estimator, KSG, fail, remains open.

As noted, unless strong inductive biases can be exploited (as illustrated for CCA), estimation of high MI is an important challenge. An interesting avenue towards developing new estimators can be by incorporating a stronger inductive bias, which could also take a form of development of better data normalization strategies and, in case of neural estimators, critic architectures. Better normalization strategies may make the estimators more invariant to considered transformations.

**Acknowledgments and Disclosure of Funding**

We would like to thank Craig Hamilton for the advice on scientific writing and help with the abstract. FG was supported by the Norwegian Financial Mechanism GRIEG-1 grant operated by Narodowe Centrum Nauki (National Science Centre, Poland) 2019/34/H/NZ6/00699. PC and AM were supported by a fellowship from the ETH AI Center.

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

# Appendix

**Table of Contents**

# A   Invariance to Continuous Injective Mappings

In this section, we provide a proof for Theorem 2.1, formally proving the invariance of mutual information with respect to continuous injective mappings. First, we recall the standard definitions from Chapters 1 and 2 of Pinsker and Feinstein [1964].

**Definition A.1.** Let $(\Omega, \mathcal{A})$ be a measurable space. A family of sets $\mathcal{E} \subseteq \mathcal{A}$ is a partition of $\Omega$ if for every two distinct $E_1$ and $E_2$ in $\mathcal{E}$ it holds that $E_1 \cap E_2 = \varnothing$ and that $\bigcup \mathcal{E} = \Omega$. If $\mathcal{E}$ is finite, we call it a finite partition.

**Definition A.2** (Mutual Information)**.** Mutual information between r.v. $X$ and $Y$ is defined as

$$\mathbf{I}(X;Y) = \sup_{\mathcal{E},\mathcal{F}} \sum_{i,j} P_{XY}(E_i \times F_j) \log \frac{P_{XY}(E_i \times F_j)}{P_X(E_i)P_Y(F_j)},$$

where the supremum is taken over all finite partitions $\mathcal{E} = \{E_1, E_2, \dots\}$ of $\mathcal{X}$ and $\mathcal{F} = \{F_1, F_2, \dots\}$ of $\mathcal{Y}$ with the usual conventions $0 \log 0 = 0$ and $0 \log 0/0 = 0$.

*Remark* A.3. If $P_{XY}$ is absolutely continuous with respect to $P_X \otimes P_Y$ (e.g., $X$ and $Y$ are finite r.v.), mutual information is equal to the following Kullback–Leibler divergence:

$$\mathbf{I}(X;Y) = \mathbf{D}_{\mathrm{KL}}\left(P_{XY} \parallel P_X \otimes P_Y\right) = \int \log f \, \mathrm{d}P_{XY},$$

where $f$ is the Radon–Nikodym derivative $f = \mathrm{d}P_{XY}/\mathrm{d}(P_X \otimes P_Y)$. If the absolute continuity does not hold, then $\mathbf{I}(X;Y) = +\infty$ [Pinsker and Feinstein, 1964, Theorem 2.1.2].

**Theorem 2.1.** *Let $\mathcal{X}$, $\mathcal{X}'$, $\mathcal{Y}$ and $\mathcal{Y}'$ be standard Borel spaces (e.g., smooth manifolds with their Borel $\sigma$-algebras) and $f \colon \mathcal{X} \to \mathcal{X}'$ and $g \colon \mathcal{Y} \to \mathcal{Y}'$ be continuous injective mappings. Then, for every $\mathcal{X}$-valued r.v. $X$ and $\mathcal{Y}$-valued r.v. $Y$ it holds that $\mathbf{I}(X;Y) = \mathbf{I}(f(X); g(Y))$.*

*Proof.* First, note that a continuous mapping between Borel spaces is measurable, so random variables $f(X)$ and $g(Y)$ are well-defined.

Then, observe that it suffices to prove that

$$\mathbf{I}(f(X); Y) = \mathbf{I}(X;Y).$$

It is well-known that we have $\mathbf{I}(f(X); Y) \leq \mathbf{I}(X;Y)$ for every measurable mapping $f$, so we will need to prove the reverse inequality.

Recall form Definition A.2 that

$$\mathbf{I}(X;Y) = \sup_{\mathcal{E},\mathcal{F}} \sum_{i,j} P_{XY}(E_i \times F_j) \log \frac{P_{XY}(E_i \times F_j)}{P_X(E_i)P_Y(F_j)},$$

where the supremum is taken over all finite partitions $\mathcal{E}$ of $\mathcal{X}$ and $\mathcal{F}$ of $\mathcal{Y}$.

The proof will go as follows: for every finite partition $\mathcal{E}$ of $\mathcal{X}$ we will construct a finite partition of $\mathcal{X}'$, which (together with the same arbitrary partition $\mathcal{F}$ of $\mathcal{Y}$) will give the same value of the sum. This will prove that $\mathbf{I}(f(X); Y) \geq \mathbf{I}(X;Y)$.

Take any finite partitions $\{E_1, \dots, E_n\}$ of $\mathcal{X}$ and $\{F_1, \dots, F_m\}$ of $\mathcal{Y}$. First, we will show that sets

$$\begin{aligned}
G_0 &= \mathcal{X}' \setminus f(\mathcal{X}) \\
G_1 &= f(E_1) \\
G_2 &= f(E_2) \\
&\;\;\vdots
\end{aligned}$$

form a partition of $\mathcal{X}'$.

It is easy to see that they are pairwise disjoint as $f$ is injective and it is obvious that their union is the whole $\mathcal{X}'$. Hence, we only need to show that they are all Borel.

For $i \geq 1$ the Lusin–Suslin theorem [Kechris, 2012, Theorem 15.1] guarantees that $G_i$ is Borel (continuous injective mappings between Polish spaces map Borel sets to Borel sets). Similarly, $f(\mathcal{X})$ is Borel and hence $G_0 = \mathcal{X}' \setminus f(\mathcal{X})$ is Borel as well.

Next, we will prove that

$$\sum_{i=1}^{n}\sum_{j=1}^{m} P_{XY}(E_i \times F_j) \log \frac{P_{XY}(E_i \times F_j)}{P_X(E_i)P_Y(F_j)} = \sum_{i=0}^{n}\sum_{j=1}^{m} P_{f(X)Y}(G_i \times F_j) \log \frac{P_{f(X)Y}(G_i \times F_j)}{P_{f(X)}(G_i)P_Y(F_j)}.$$

Note that $P_{f(X)}(G_0) = P_X(f^{-1}(G_0)) = P_X(\varnothing) = 0$, so that we can ignore the terms with $i = 0$.
For $i \geq 1$ we have

$$P_{f(X)}(G_i) = P_X(f^{-1}(G_i)) = P_X(f^{-1}(f(E_i))) = P_X(E_i)$$

as $f$ is injective.

Similarly, $f \times 1_{\mathcal{Y}}$ is injective and maps $E_i \times F_j$ onto $G_i \times F_j$, so

$$P_{f(X)Y}(G_i \times F_j) = P_{XY}\big((f \times 1_{\mathcal{Y}})^{-1}(G_i \times F_j)\big) = P_{XY}(E_i \times F_j).$$

This finishes the proof. $\qquad\square$

## B  Spiral Diffeomorphisms

In this section we will formally define the spiral diffeomorphisms and prove that they preserve the centred Gaussian distribution $\mathcal{N}(0, \sigma^2 I_n)$. They were inspired by the "swirling construction", discussed by Hauberg [2019].

**Definition B.1.** Let $O(n)$ be the orthogonal group and $\gamma \colon \mathbb{R} \to O(n)$ be any smooth function. The mapping

$$f \colon \mathbb{R}^n \to \mathbb{R}^n$$

given by $f(x) = \gamma(||x||^2)x$ will be called the spiral.

**Example B.2.** *Let $\mathfrak{so}(n)$ be the vector space of all skew-symmetric matrices. For every two matrices $R \in O(n)$ and $A \in \mathfrak{so}(n)$ and any smooth mapping $g \colon \mathbb{R} \to \mathbb{R}$ we can form a path*

$$\gamma(t) = R\exp(g(t)A).$$

*In particular, $R = I_n$ and $g(t) = vt$ yields the spirals defined earlier.*

From the above definition it is easy to prove that spirals are diffeomorphisms. We formalize it by the following lemma.

**Lemma B.3.** *Every spiral is a diffeomorphism.*

*Proof.* Let $\gamma$ be the path leading to the spiral $f$. As matrix inverse is a smooth function, we can define a smooth function $h(x) = \gamma\left(||x||^2\right)^{-1} x$.

Note that $||f(x)||^2 = ||x||^2$, as orthogonal transformation $\gamma\left(||x||^2\right)$ preserves distances.

Then

$$\begin{aligned}
h(f(x)) &= \gamma\left(||f(x)||^2\right)^{-1} f(x) \\
&= \gamma\left(||x||^2\right)^{-1} f(x) \\
&= \gamma\left(||x||^2\right)^{-1} \gamma\left(||x||^2\right) x \\
&= x.
\end{aligned}$$

Analogously, we prove that $f \circ h$ also is the identity mapping. $\qquad\square$

Finally, we will give the proof that spirals leave the centred normal distribution $\mathcal{N}(0, \sigma^2 I_n)$ invariant. This will follow from a slightly stronger theorem:

**Theorem B.4.** *Let $P$ be a probability measure on $\mathbb{R}^n$ with a smooth probability density function $p$ which factorizes as:*

$$p(x) = u\left(||x||^2\right)$$

*for some smooth function $u \colon \mathbb{R} \to \mathbb{R}$.*

*Then, every spiral $f \colon \mathbb{R}^n \to \mathbb{R}^n$ preserves the measure $P$:*

$$f_\# P = P,$$

*where $f_\# P = P \circ f^{-1}$ is the pushforward measure.*

*In particular, for $P = \mathcal{N}(0, \sigma^2 I_n)$ (a centered isotropic Gaussian distribution), if $X \sim \mathcal{N}(0, \sigma^2 I_n)$, then $f(X) \sim \mathcal{N}(0, \sigma^2 I_n)$.*

*Proof.* As we have already observed, for every $x \in \mathbb{R}^n$ it holds that $||f(x)||^2 = ||x||^2$, so the PDF evaluated at both points is equal

$$p(x) = p(f(x))$$

Hence, we need to prove that for every point $x \in \mathbb{R}^n$, $|\det f'(x)| = 1$, where $f'(x)$ is the Jacobian matrix of $f$ evaluated at the point $x$.

The proof of this identity was given by Carapetis [2022], but for the sake of completeness we supply a variant of it below.

First, we calculate the Jacobian matrix:

$$\frac{\partial f_i(x)}{\partial x_j} = \frac{\partial}{\partial x_j} \sum_k \gamma(||x||^2)_{ik} x_k$$

$$= \sum_k \gamma(||x||^2)_{ik} \frac{\partial x_k}{\partial x_j} + \sum_k x_k \frac{\partial \gamma(||x||^2)_{ik}}{\partial x_j}$$

$$= \gamma(||x||^2)_{ij} + \sum_k x_k \cdot 2x_j \cdot \gamma'(||x||^2)_{ik}$$

$$= \gamma(||x||^2)_{ij} + 2 \left( \sum_k \gamma'(||x||^2)_{ik} x_k \right) x_j$$

Hence,

$$f'(x) = \gamma(||x||^2) + 2\gamma'(||x||^2) x x^T.$$

Now fix point $x$ and consider the matrix $\gamma(||x||^2) \in O(n)$. Define a mapping $y \mapsto \gamma(||x||^2)^{-1} f(y)$. As we have already observed, multiplication by $\gamma(||x||^2)^{-1}$ does not change the value of the PDF. It also cannot change the $|\det f'(x)|$ value as because $\gamma(||x||^2)^{-1}$ is a linear mapping, its derivative is just the mapping itself and because $\gamma(||x||^2)^{-1}$ is an orthogonal matrix, we have $|\det \gamma(||x||^2)^{-1}| = 1$. Hence, without loss of generality we can assume that $\gamma(||x||^2) = I_n$ and

$$f'(x) = I_n + 2Axx^T,$$

where $A := \gamma'(||x||^2) \in \mathfrak{so}(n)$ is a skew-symmetric matrix.

If $x = 0$ we have $|\det f'(x)| = |\det I_n| = 1$. In the other case, we will work in coordinates induced by an orthonormal basis $e_1, \ldots, e_n$ in which the first basis vector is $e_1 = x/||x||$. Hence,

$$f'(x) = I_n + 2||x||^2 A e_1 e_1^T.$$

For $i \geq 2$ we have $e_1^T e_i = 0$ and consequently

$$f'(x) e_i = e_i + 2||x||^2 A e_1 e_1^T e_i = e_i.$$

This shows that in this basis we have

$$f'(x) = \begin{pmatrix} a_1 & 0 & 0 & \ldots & 0 \\ a_2 & 1 & 0 & \ldots & 0 \\ a_3 & 0 & 1 & \ldots & 0 \\ \vdots & \vdots & \vdots & \ddots & 0 \\ a_n & 0 & 0 & \cdots & 1 \end{pmatrix},$$

where the block on the bottom right is the identity matrix $I_{n-1}$ and $\det f'(x) = a_1$.

The first column is formed by the coordinates of

$$f'(x)e_1 = e_1 + 2||x||^2 Ae_1$$

and we have

$$a_1 = e_1^T f'(x)e_1 = 1 + 2||x||^2 e_1^T Ae_1 = 1 + 0 = 1$$

as $A \in \mathfrak{so}(n)$ is skew-symmetric. $\qquad\square$

## C  Overview of Estimators

There exists a broad literature on (conditional) mutual information estimators ranging from estimators on discrete data [Paninski, 2003, Valiant and Valiant, 2011, Vinh et al., 2014, Marx and Vreeken, 2019], over estimators for continuous data [Kraskov et al., 2004, Darbellay and Vajda, 1999, Belghazi et al., 2018] to estimators that can consider discrete-continuous mixture data [Gao et al., 2017a, Marx et al., 2021, Mesner and Shalizi, 2020]. Here, we focus on the most commonly used class, which also finds applications in representation learning: estimators for continuous data. For continuous data, the most widely propagated classes of estimators are histogram or partition-based estimators, nearest neighbor-based density estimators, kernel methods, and neural estimators. In the following, we briefly discuss a representative selection of estimators from these classes, as well as model-based estimation.

**Histogram-Based Estimators**  A natural way to approximate mutual information, as stated in Definition A.2, is by partitioning the domain space of $X$ and $Y$ into segments and estimate the densities locally. This approach is also known as histogram density estimation, where the probability mass function is approximated locally for each hyper-rectangle $E_i$ by $\hat{P}_X(E_i) = \frac{n_i/N}{\text{Vol}_i}$, where $\text{Vol}_i$ is the volume of $E_i$, and $n_i$ refers to the number of data points residing in $E_i$.

To estimate mutual information, we explore the space of allowed partitionings and select the partitioning that maximizes mutual information [Darbellay and Vajda, 1999, Cellucci et al., 2005]

$$E_{\text{Disc},N}^{XY} = \sup_{\mathcal{E},\mathcal{F}} \sum_{i,j} \hat{P}_{XY}(E_i \times F_j) \log \frac{\hat{P}_{XY}(E_i \times F_j)}{\hat{P}_X(E_i)\hat{P}_Y(F_j)} .$$

Many estimators follow this idea. Darbellay and Vajda [1999] use adaptive partitioning trees and propose to split bins until the density within a bin is uniform, which they check with the $\chi^2$ test. Daub et al. [2004] use B-Splines to assign data points to bins, and other authors use regularization based on the degrees of freedom of partitioning to avoid overfitting [Suzuki, 2016, Cabeli et al., 2020, Vinh et al., 2014, Marx et al., 2021].

A well-known property of histogram-based estimators is that they are consistent for $N \to \infty$ if the volume of the largest hypercube goes to zero, while the number of hypercubes grows sub-linearly with the number of samples, $N$ (see e.g. Cover and Thomas [2006]). For low-dimensional settings, histogram density estimation can provide accurate results, but the strategy suffers from the curse of dimensionality [Pizer et al., 1987, Klemelä, 2009].

In our evaluation, we resort to partitioning estimators based on a fixed number of grids (10 equal-width bins per dimension), since more recent partitioning-based estimators focus on estimating conditional MI and only allow $X$ and $Y$ to be one-dimensional [Suzuki, 2016, Marx et al., 2021, Cabeli et al., 2020], or do not have a publicly available implementation [Darbellay and Vajda, 1999].

***k*NN Estimators**  An alternative to histogram-based methods is to use $k$-nearest neighbor ($k$NN) distances to approximate densities locally. Most generally, $k$NN estimators express mutual information as the sum $H(X) + H(Y) - H(X,Y)$, which can be done under the assumptions discussed in Remark A.3, and estimate the individual entropy terms as

$$\hat{H}((x_1,\ldots,x_N)) = -\frac{1}{N}\sum_{i=1}^{N} \log \frac{k(x_i)/N}{\text{Vol}_i} ,$$

where $k(x_i)$ is the number of data points other than $x_i$ in a neighborhood of $x_i$, and $\text{Vol}_i$ is the volume of the neighborhood [Kozachenko and Leonenko, 1987]. The quality of the estimator depends on two

factors: i) the distance metric used to estimate the nearest neighbor distances, and ii) the estimation of the volume terms.

The most well-known estimator is the Kraskov, Stögbauer and Grassberger (KSG) estimator [Kraskov et al., 2004]. In contrast to the first $k$NN-based estimators [Kozachenko and Leonenko, 1987], Kraskov et al. [2004] propose to estimate the distance to the $k$-th neighbor $\rho_{i,k}/2$ for each data point $i$ only on the joint space of $\mathcal{Z} = \mathcal{X} \times \mathcal{Y}$, with

$$D_Z(z_i, z_j) = \max\{D_X(x_i, x_j), D_Y(y_i, y_j)\},$$

where $D_X$ and $D_Y$ can refer to any norm-induced metric (e.g., Euclidean $\ell_2$, Manhattan $\ell_1$ or Chebyshev $\ell_\infty$).[5] To compute the entropy terms of the sub-spaces $\mathcal{X}$ and $\mathcal{Y}$, we count the number of data points with distance smaller than $\rho_{i,k}/2$, i.e. we define $n_{x,i}$ as

$$n_{x,i} = \left| \left\{ x_j \, : \, D_X(x_i, x_j) < \frac{\rho_{i,k}}{2}, i \neq j \right\} \right|, \tag{1}$$

and specify $n_{y,i}$ accordingly. As a consequence, all volume-related terms cancel and the KSG estimator reduces to

$$E_{\text{KSG},N}^{XY} = \psi(k) + \psi(n) - \frac{1}{n} \sum_{i=1}^{n} \psi(n_{x,i}+1) + \psi(n_{y,i}+1). \tag{2}$$

The diagamma function[6] $\psi$ was already introduced by Kozachenko and Leonenko [1987] to correct for biases induced by the volume estimation. For a detailed exposition of the issue, we refer to Kraskov et al. [2004, Sec. II.A].

There exist various extensions of the classical KSG estimator. When selecting the $L_\infty$ norm as a distance measure for $D_X$ and $D_Y$, $E_{\text{KSG},N}^{XY}$ can be efficiently computed with the K-d trie method [Vejmelka and Hlaváčková-Schindler, 2007]. Other approaches aim to improve the estimation quality by explicitly computing the volume terms via SVD [Lord et al., 2018] or PCA [Lu and Peltonen, 2020], or use manifold learning [Marx and Fischer, 2022] or normalizing flows [Ao and Li, 2022] to estimate the nearest neighbors more efficiently in high-dimensional settings.

**Kernel Estimators**  A smaller class of estimators builds upon kernel density estimation (KDE) [Moon et al., 1995, Paninski and Yajima, 2008]. Classical approaches compute the density functions for the individual entropy terms via kernel density estimation, which is well-justified under the assumption of $P_{XY}$ being absolutely continuous with respect to $P_X \otimes P_Y$ (cf. Remark A.3). i.e., the density for a $d$-dimensional random vector $X$ is approximated as

$$\hat{p}(x) = \frac{1}{Nh^d} \sum_{i=1}^{N} K\left(\frac{x - x_i}{h}\right),$$

where $h$ is the smoothing parameter or bandwidth of a Gaussian kernel [Silverman, 2018].

Moon et al. [1995] propose to estimate mutual information using a Gaussian kernel, for which the bandwidth can be estimated optimally [Silverman, 2018]. Similar to histogram-based approaches, KDE-based estimators with fixed bandwidth $h$ are consistent if $Nh^d \to \infty$ as $h \to 0$.

A more recent proposal, LNN [Gao et al., 2017b], uses a data-dependent bandwidth based on nearest-neighbor distances, to avoid fixing $h$ globally.

**Neural Estimators**  Neural network approaches such as MINE [Belghazi et al., 2018] use a variational lower bound on mutual information using the Donsker–Varadhan theorem: let $f \colon \mathcal{X} \times \mathcal{Y} \to \mathbb{R}$ be a bounded measurable function and

$$V_f(X, Y) = \sup_f \mathbb{E}_{P_{XY}}[f(X, Y)] - \log \mathbb{E}_{P_X \otimes P_Y}[\exp f(X, Y)].$$

Under sufficient regularity conditions on $P_{XY}$, $\mathbf{I}(X; Y)$ is equal to $V_f(X, Y)$, which is evaluated over all bounded measurable mappings $f$. In practice, the right hand side is optimized with respect

---

[5]In the original approach, the $L_\infty$ is used, and Gao et al. [2018] derives theoretical properties of the estimator for the $L_2$ norm.

[6]The digamma function is defined as $\psi(x) = \mathrm{d} \log \Gamma(x)/\mathrm{d}x$. It satisfies the recursion $\psi(x+1) = \psi(x) + \frac{1}{x}$ with $\psi(1) = -\gamma$, where $\gamma \approx 0.577$ is the Euler–Mascheroni constant.

to the parameters of a given parametric predictive model $f$, as a neural network. Approaches following this idea also count as discriminative approaches since the neural network is trained to distinguish points from the joint and marginal distribution. The literature on variational lower bounds of mutual information is growing [Oord et al., 2018, Song and Ermon, 2020] and the idea of using a discriminative approach to estimate mutual information has also been explored by Koeman and Heskes [2014], who used random forests. For an excellent theoretical overview of different neural estimators we refer to Poole et al. [2019].

**Model-based Estimators**  Finally, if one assumes that $P_{XY}$ is a member of a family of distributions $Q_{XY}(\theta)$ with known mutual information, one can estimate $\theta$ basing on the data $((x_1, y_1), \ldots, (x_N, y_N))$ and calculate the corresponding mutual information. For example, if one assumes that there exist jointly multivariate normal variables $\tilde{X} = (\tilde{X}_1, \ldots, \tilde{X}_r, E_1, \ldots, E_{m-r})$ and $\tilde{Y} = (\tilde{Y}_1, \ldots, \tilde{Y}_r, \tilde{E}_1, \ldots, \tilde{E}_{n-r})$ such that the cross-covariance terms are zero apart from $\text{Cov}(\tilde{X}_i, \tilde{Y}_i)$, and that $X = A\tilde{X} + a$ and $Y = B\tilde{Y} + b$ for invertible matrices $A$ and $B$ and vectors $a \in \mathbb{R}^m$ and $b \in \mathbb{R}^n$, then the mutual information can be found as

$$\mathbf{I}(X;Y) = -\sum_{i=1}^{r} \log \sqrt{1 - \text{Cor}(\tilde{X}_r, \tilde{Y}_r)^2}.$$

The model parameters $A$, $B$, $a$, and $b$, as well as the pairs of corresponding r.v. $\tilde{X}_r$ and $\tilde{Y}_r$, can be found by maximising likelihood using canonical correlation analysis (CCA) [Murphy, 2023, Ch. 28], which has been previously explored by Kay [1992].

# D  Additional Information on Benchmark Tasks

## D.1  Task Descriptions

**Bimodal Variables**  Consider a r.v. $X$ distributed according to a Gaussian mixture model with CDF $F = w_1 F_1 + w_2 F_2$, where $F_1$ and $F_2$ are CDFs of two Gaussian distributions and $w_1$ and $w_2$ are positive weights constrained as $w_1 + w_2 = 1$. The CDF is everywhere positive, continuous, and strictly increasing. Hence, it has an inverse, the quantile function, $F^{-1} \colon (0, 1) \to \mathbb{R}$, which is continuous and injective as well (see Lemma D.1). Thus, we can define r.v. $X' = F^{-1}(X)$, where $X \sim \text{Uniform}(0, 1)$ and $\mathbf{I}(X;Y) = \mathbf{I}(X';Y)$ for every r.v. $Y$. We generated two CDFs of Gaussian mixtures and implemented a numerical inversion in JAX [Bradbury et al., 2018] defining both $X'$ and $Y'$ to follow a bimodal distribution.

For the experimental values, we used

$$F_X(x) = 0.3F(x) + 0.7F(x - 5)$$
$$F_Y(y) = 0.5F(x + 1) + 0.5F(x - 3),$$

where $F$ is the CDF of the standard normal distribution $\mathcal{N}(0, 1)$.

**Wiggly Mapping**  As the "wiggly mapping" we used

$$w_X(x) = x + 0.4\sin x + 0.2\sin(1.7x + 1) + 0.03\sin(3.3x - 2.5)$$
$$w_Y(y) = y - 0.4\sin(0.4y) + 0.17\sin(1.3y + 3.5) + 0.02\sin(4.3y - 2.5)$$

and we used visualisations to make sure that they do not heavily distort the distribution.

**Additive Noise**  For the additive noise model of $Y = X + N$ where $X \sim \text{Uniform}(0, 1)$ and $N \sim \text{Uniform}(-\varepsilon, \varepsilon)$ it holds that

$$\mathbf{I}(X;Y) = \begin{cases} \varepsilon - \log(2\varepsilon) & \text{if } \varepsilon \leq 1/2 \\ (4\varepsilon)^{-1} & \text{otherwise} \end{cases}$$

implying that $\mathbf{I}(X;Y) = 1.7$ for $\varepsilon = 0.1$ and $\mathbf{I}(X;Y) = 0.3$ for $\varepsilon = 0.75$.

**Swiss Roll Embedding**   Consider a bivariate distribution $P_{XY}$ with uniform margins $X \sim$ Uniform$(0,1)$ and $Y \sim$ Uniform$(0,1)$ and $\mathbf{I}(X;Y) = 0.8$. The Swiss roll mapping is an embedding $i\colon (0,1)^2 \to \mathbb{R}^3$ given by $i(x,y) = (e(x), y)$, where $e\colon (0,1) \to \mathbb{R}^2$ is given by

$$e(x) = \frac{1}{21}(t(x)\cos t(x), t(x)\sin t(x)), \qquad t(x) = \frac{3\pi}{2}(1+2x).$$

Note that $P_{X'Y'}$ does not have a PDF with respect to the Lebesgue measure on $\mathbb{R}^3$. We visualise the Swiss roll distribution in Fig. 1.

**Spiral Diffeomorphism**   For the precise definition and properties of the general spiral diffeomorphism consult Appendix B.

In our benchmark we applied the spiral $x \mapsto \exp(vA\|x\|^2)x$ to both $X$ and $Y$ variables. We used an $m \times m$ skew-symmetric matrix $A_X$ with $(A_X)_{12} = -(A_X)_{21} = 1$ and an $n \times n$ skew-symmetric matrix $A_Y$ with $(A_Y)_{23} = -(A_Y)_{32} = 1$. Each of these effectively "mixes" only two dimensions.

The speed parameters $v_X$ and $v_Y$ were chosen as $v_X = 1/m$ and $v_Y = 1/n$ to partially compensate for the fact that if $X \sim \mathcal{N}(0, I_m)$, then $\|X\|^2 \sim \chi_m^2$ and has mean $m$.

**Multivariate Student**   Let $\Omega$ be an $(m+n) \times (m+n)$ positive definite dispersion matrix and $\nu$ be a positive integer (the degrees of freedom). By sampling an $(m+n)$-dimensional random vector $(\tilde{X}, \tilde{Y}) \sim \mathcal{N}(0, \Omega)$ and a random scalar $U \sim \chi_\nu^2$ one can define rescaled variables $X = \tilde{X}\sqrt{\nu/U}$ and $Y = \tilde{Y}\sqrt{\nu/U}$, which are distributed according to the multivariate Student distribution. For $\nu = 1$ this reduces to the multivariate Cauchy distribution (and the first two moments are not defined), for $\nu = 2$ the mean exists, but covariance does not, and for $\nu > 2$ both the mean and covariance exist, with $\mathrm{Cov}(X,Y) = \frac{\nu}{\nu-2}\Omega$, so that the tail behaviour can be controlled by changing $\nu$. In particular, for $\nu \gg 1$ because of the concentration of measure phenomenon $U$ has most of its probability mass around $\nu$ and the variables $(X,Y)$ can be well approximated by $(\tilde{X}, \tilde{Y})$.

Arellano-Valle et al. [2013] proved that $\mathbf{I}(X;Y)$ can be computed via the sum of the mutual information of the Gaussian distributed basis variables and a correction term, i.e., $\mathbf{I}(X;Y) = \mathbf{I}(\tilde{X}; \tilde{Y}) + c(\nu, m, n)$, where

$$c(\nu, m, n) = f(\nu) + f(\nu+m+n) - f(\nu+m) - f(\nu+n), \quad f(x) = \log\Gamma\left(\frac{x}{2}\right) - \frac{x}{2}\psi\left(\frac{x}{2}\right),$$

and $\psi$ is the digamma function.

Contrary to the Gaussian case, even for $\Omega = I_{m+n}$, the mutual information $\mathbf{I}(X;Y) = c(\nu, m, n)$ is non-zero, as $U$ provides some information about the magnitude. In the benchmark we use this dispersion matrix to quantify how well the estimators can use the information contained in the tails, rather than focusing on estimating the Gaussian term.

### D.2   Covariance Matrix Family

We generated jointly multivariate normal variables $X = (X_1, \ldots, X_m)$ and $Y = (Y_1, \ldots, Y_n)$ with the following procedure.

Consider i.i.d. Gaussian variables

$$U_{\mathrm{all}}, U_X, U_Y, Z_1, \ldots, Z_K, E_1 \ldots, E_m, F_1, \ldots, F_n, E'_{m-K}, \ldots, E'_m, F'_{n-K}, \ldots, F'_n \sim \mathcal{N}(0,1).$$

Now let $\varepsilon_X, \varepsilon_Y, \eta_X, \eta_Y, \alpha, \beta_X, \beta_Y, \lambda \in \mathbb{R}$ be hyperparameters and for $l = 1, 2, \ldots, K$ define

$$X_l = \varepsilon_X E_l + \alpha U_{\mathrm{all}} + \beta_X U_X + \lambda Z_l, \quad Y_l = \varepsilon_Y F_l + \alpha U_{\mathrm{all}} + \beta_Y U_Y + \lambda Z_l,$$

for $l = K+1, \ldots, m$ define

$$X_l = \varepsilon_X E_l + \alpha U_{\mathrm{all}} + \beta_X U_X + \eta_X E'_l$$

and for $l = K+1, \ldots, n$ define

$$Y_l = \varepsilon_Y F_l + \alpha U_{\mathrm{all}} + \beta_Y U_Y + \eta_Y F'_l.$$

The interpretation of the hyperparameters is the following:

1. $\alpha$ contributes to the correlations between all the possible pairs of variables.

2. $\beta_X$ contributes to the correlations between the $X$ variables.

3. $\beta_Y$ contributes to the correlations between the $Y$ variables.

4. $\lambda$ gives additional "interaction strength" between pairs of variables $X_l$ and $Y_l$ for $l = 1, \ldots, K$.

5. $\varepsilon_X$ and $\varepsilon_Y$ control part of the variance non-shared with any other variable.

6. $\eta_X$ and $\eta_Y$ can be used to increase the variance of $X_l$ and $Y_l$ for $l > K$ to match the variance of variables $l \leq K$ due to the $\lambda$.

Every term of the covariance matrix is easy to calculate analytically:

$$\text{Cov}(X_i, X_j) = \alpha^2 + \beta_X^2 + \mathbf{1}[i = j]\big(\varepsilon_X^2 + \lambda^2 \mathbf{1}[i \leq K] + \eta_X^2 \mathbf{1}[i > K]\big)$$

$$\text{Cov}(Y_i, Y_j) = \alpha^2 + \beta_Y^2 + \mathbf{1}[i = j]\big(\varepsilon_Y^2 + \lambda^2 \mathbf{1}[i \leq K] + \eta_Y^2 \mathbf{1}[i > K]\big)$$

$$\text{Cov}(X_i, Y_j) = \alpha^2 + \lambda^2 \mathbf{1}[i = j] \mathbf{1}[i \leq K]$$

In the following, we analyse some special cases.

**Dense Interactions**  Consider $\lambda = \eta_X = \eta_Y = \beta_X = \beta_Y = 0$ and write $\varepsilon := \varepsilon_X = \varepsilon_Y$. We have

$$\text{Cov}(X_i, X_j) = \text{Cov}(Y_i, Y_j) = \alpha^2 + \varepsilon^2 \mathbf{1}[i = j]$$

$$\text{Cov}(Y_i, Y_j) = \alpha^2$$

Hence, between every pair of distinct variables we have the same correlation $\frac{\alpha^2}{\alpha^2 + \varepsilon^2}$.

**Sparse Interactions**  Consider $\alpha = 0$, $\varepsilon := \varepsilon_X = \varepsilon_Y$, $\eta_X = \eta_Y = \lambda$ and $\beta_X = \beta_Y = 0$

$$\text{Cov}(X_i, X_j) = \text{Cov}(Y_i, Y_j) = \mathbf{1}[i = j](\varepsilon^2 + \lambda^2)$$

$$\text{Cov}(X_i, Y_j) = \lambda^2 \mathbf{1}[i = j] \mathbf{1}[i \leq K]$$

All the variables have the same variance, but the covariance structure is now different. Between (distinct) $X_i$ and $X_j$ the correlation is zero and similarly for (distinct) $Y_i$ and $Y_j$.

Correlations between $X_i$ and $Y_j$ are zero unless $i = j \leq K$, when $\text{Cor}(X_i, Y_i) = \lambda^2/(\varepsilon^2 + \lambda^2)$.

**Interpolation**  In Section 4 we discussed a two-stage interpolation from dense matrices to sparse matrices: in the first stage $\alpha$ is decreased increasing $\lambda$ at the same time to keep the mutual information constant. In the second stage we decrease the number of interacting variables $K$ increasing $\lambda$. The selected covariance matrices from this two-stage process are visualised in Fig. 7.

### D.3   Technical Results

In this subsection we add the technical results, related to the copula theory [Nelsen, 2007] and uniformization of random variables.

**Lemma D.1.**  *Let $X$ be a real-valued r.v. with a strictly increasing, positive everywhere, and continuous CDF $F$. Then*

$$F \colon \mathbb{R} \to (0, 1)$$

*is a homeomorphism.*

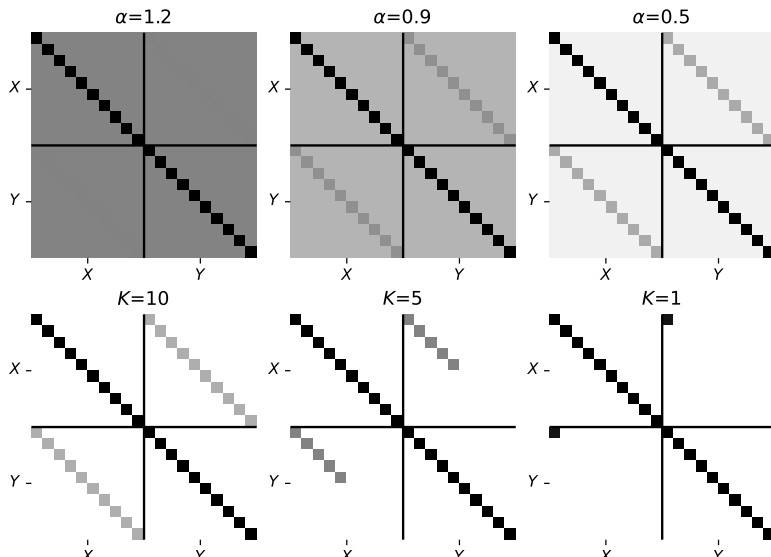

Figure 7: Interpolation between dense and sparse covariance matrices.

*Proof.* As $F$ is a CDF we have $F(\mathbb{R}) \subseteq [0, 1]$. Note that $0$ is excluded from the image as $F$ is positive everywhere and $1$ is excluded as $F$ is strictly increasing. Hence, $F(\mathbb{R}) \subseteq (0, 1)$. Using continuity, strict monotonicity, and

$$\lim_{x \to -\infty} F(x) = 0, \qquad \lim_{x \to +\infty} F(x) = 1$$

we note that $F$ is a bijection onto $(0, 1)$. Analogously, we prove that

$$F((a, b)) = (F(a), F(b))$$

for all $a < b$, so $F$ is an open map. As a continuous bijective open map $F$ is a homeomorphism. $\qquad\square$

**Lemma D.2.** *Let $X$ be a real-valued r.v. with a CDF $F$ which is continuous, strictly increasing and positive everywhere. Then, $F(X)$ is uniformly distributed on $(0, 1)$.*

*Proof.* Our assumptions on $F$ imply that it is a continuous homeomorphism $\mathbb{R} \to (0, 1)$.

Let now $y \in (0, 1)$. We have

$$P(F(X) \le y) = P(X \le F^{-1}(y)) = F\left(F^{-1}(y)\right) = y.$$

$\qquad\square$

# E  Additional Experiments

## E.1  Pointwise Mutual Information

To understand why neural estimators are not able to estimate the mutual information contained in the Student distribution, we decided to qualitatively study the learned critic.

In the NWJ approach [Nguyen et al., 2007], for $N \to \infty$ data points the optimal critic $f(x, y)$ is given by

$$f(x, y) = 1 + \text{PMI}(x, y),$$

where

$$\text{PMI}(x, y) = \log \frac{p_{XY}(x, y)}{p_X(x) p_Y(y)}$$

is the pointwise mutual information [Pinsker and Feinstein, 1964, Ch. 2] and $p_{XY}$, $p_X$ and $p_Y$ are the PDFs of measures $P_{XY}$, $P_X$ and $P_Y$, respectively.

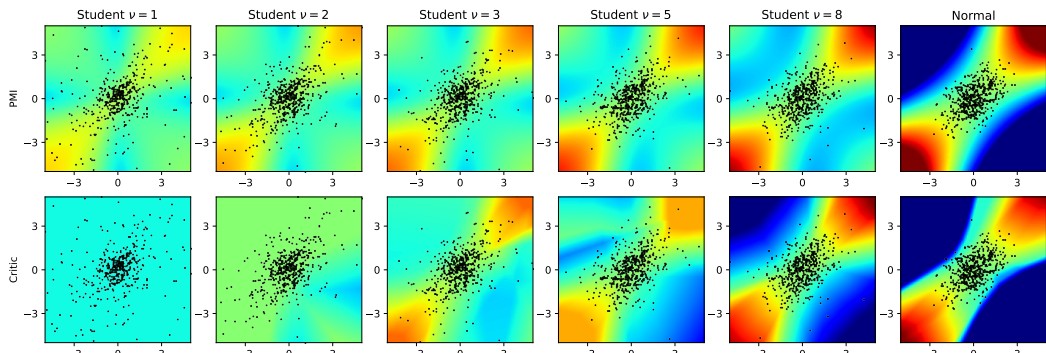

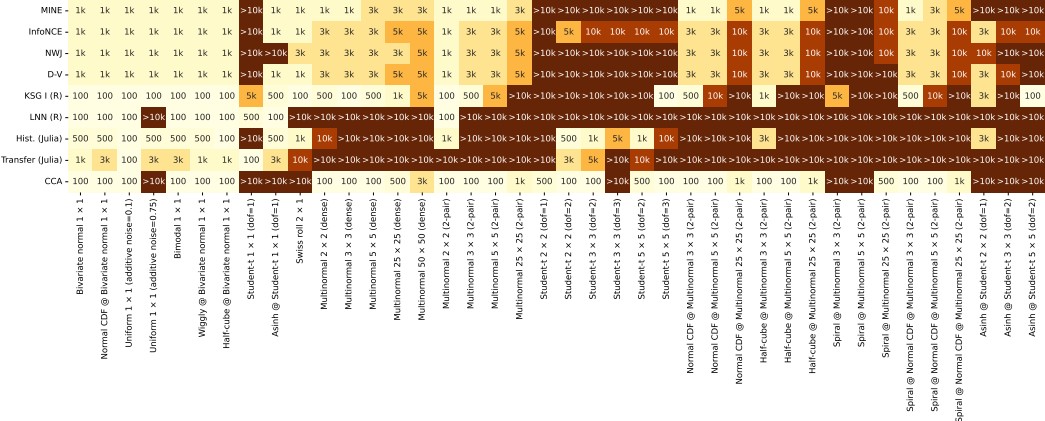

Figure 8: Qualitative comparison between ground-truth pointwise MI and learned critic for bivariate Student and normal distributions.

Figure 9: Minimal number of samples needed for the mutual information estimate to fall between $2/3$ and $3/2$ of the true MI. The minimal number of samples tested is 100, the maximal 10 000. The entry $> 10k$ means that the desired accuracy was never achieved. For neural estimators 1 000 is the minimal possible number of samples, since we use a 50% train-test split and a batch size of 256.

We trained NWJ estimator with the default hyperparameters on $N = 5\,000$ data points and visualised the critic predictions (with 1 subtracted), true PMI, and 500 samples from the distribution in Fig. 8. The data is drawn from a bivariate Student distribution with dispersion matrix

$$\Omega = \begin{pmatrix} 1 & 0.5 \\ 0.5 & 1 \end{pmatrix}$$

and varying degrees of freedom $\nu \in \{1, 2, 3, 5, 8\}$. Additionally, we repeated the experiment for the bivariate normal distribution with covariance matrix given by the dispersion, which corresponds to the limit $\nu \to \infty$.

Qualitatively we see that the critic does not learn the correct PMI for $\nu \in \{1, 2\}$, with better similarities for $\nu \in \{3, 5, 8\}$. For the normal distribution, the critic qualitatively approximates well the PMI function in the regions where the data points are available.

We hypothesise that these problems can be partly atributed to the longer tails of Student distributions (as the critic may be unable to extrapolate to the regions where few points are available) and partly to the analytical form of the PMI function — for the considered bivariate normal distribution the PMI is given by a simple quadratic function, which is not the case for the bivariate Student distribution.

### E.2   Sample Size Requirements

We investigated the minimal number of samples needed to estimate the mutual information changing the number of samples in $N \in \{100, 500, 1000, 3000, 5000, 10\,000\}$. We say that sample $\ell$ suffices

to solve the task if for every $\ell' \geq \ell$ the estimator provides the estimate between 2/3 and 3/2 of the ground-truth mutual information.

We present the minimal number of samples $\ell$ as a function of benchmark task and considered estimator in Fig. 9. Note that CCA and KSG typically need very small amount of samples to solve the task. As neural estimators use batch size of 256 and split the sample into training and test data sets, it is not possible for them to have $\ell < 1000$, which is the smallest number of samples considered greater than $2 \cdot 256 = 512$.

### E.3 Preprocessing

In Fig. 2 and Fig. 9 we transformed all samples by removing the mean and scaling by the standard deviation in each dimension calculated empirically from the sample. This strategy ensures that the scales on different axes are similar. Note that this strategy is not always suitable (e.g., if the distance metric used in the $k$NN approach should weight select dimensions differently) and makes the observed data points not i.i.d. (as the "shared" information, sample mean and variances, has been used to rescale the variables), although the effect of it seems to be negligible on most tasks in the benchmark.

Additionally, we considered two other preprocessing strategies, implemented using the `scikit-learn` package [Pedregosa et al., 2011]: making each marginal distribution uniform $\mathrm{Uniform}(0, 1)$ by calculating empirical CDF along each axis (see Section D for the technical lemmata discussing this operation) and making each marginal distribution a Gaussian variable $\mathcal{N}(0, 1)$. These approaches may affect the effective sample size more strongly than the previous rescaling strategy, as estimating empirical CDF requires more data points than estimating mean and standard deviation.

We used $N = 10\,000$ data points and rerun the benchmark (see Fig. 10). Overall, the impact of the "uniformization" strategy appeared to be insubstantial for most tasks, and in certain cases negative. The "Gaussianisation" strategy improved estimation for Student distributions and helped histogram based approaches in low-dimensional tasks. Interestingly, both approaches worsen the performance of CCA on Student distributions.

While the improvements gained from "Gaussianisation" were minor, the approach is simplistic, treating each axis separately. Ideally, appropriate preprocessing would be able to transform $P_{XY}$ into a better-behaved distribution $P_{f(X)g(Y)}$, and thus simplify the estimation. In certain cases such preprocessing could (partially) undo transformations considered in our benchmark, and in effect make the estimators invariant (or more robust) to these transformations (cf. Section 4). We consider the design of informed preprocessing strategies an important open problem.

### E.4 Hyperparameter Selection

**CCA** We used the implementation provided in the `scikit-learn` [Pedregosa et al., 2011] Python package (version `1.2.2`). We set the latent space dimension to the smaller of the dimensions of the considered random vectors $X$ and $Y$.

**Histograms** Adaptive estimators such as JIC [Suzuki, 2016] or MIIC [Cabeli et al., 2020] were not applicable to the high-dimensional data we considered, hence we resort to using an equal-width estimator implemented in `TransferEntropy` library (version `1.10.0`). We fixed the number of bins to 10, although the optimal number of bins is likely to depend on the dimensionality of the problem.

**KSG** As a default, we used the `KSG-1` variant from the `rmi` package with 10 neighbors. However, as shown in our ablation in Fig. 11, changing the number of neighbors or using the `KSG-2` variant did not considerably change the estimates. We compared the used implementation with a custom Python implementation and one from `TransferEntropy` library. The version implemented in the `TransferEntropy` suffers from strong positive bias.

**LNN** We used the implementation from the `rmi` package (version `0.1.1`) with 5 neighbors and truncation parameter set to 30. Ablations can be seen in Fig. 11.

**Neural Estimators** We split the data set into two equal-sized parts (training and test) and optimized the statistics network on the training data set using the Adam optimiser with initial learning rate set to 0.1. We used batch size of 256 and ran each algorithm for up to $10\,000$ training steps with early stopping (checked every 250 training steps). We returned the highest estimate on the test data set.

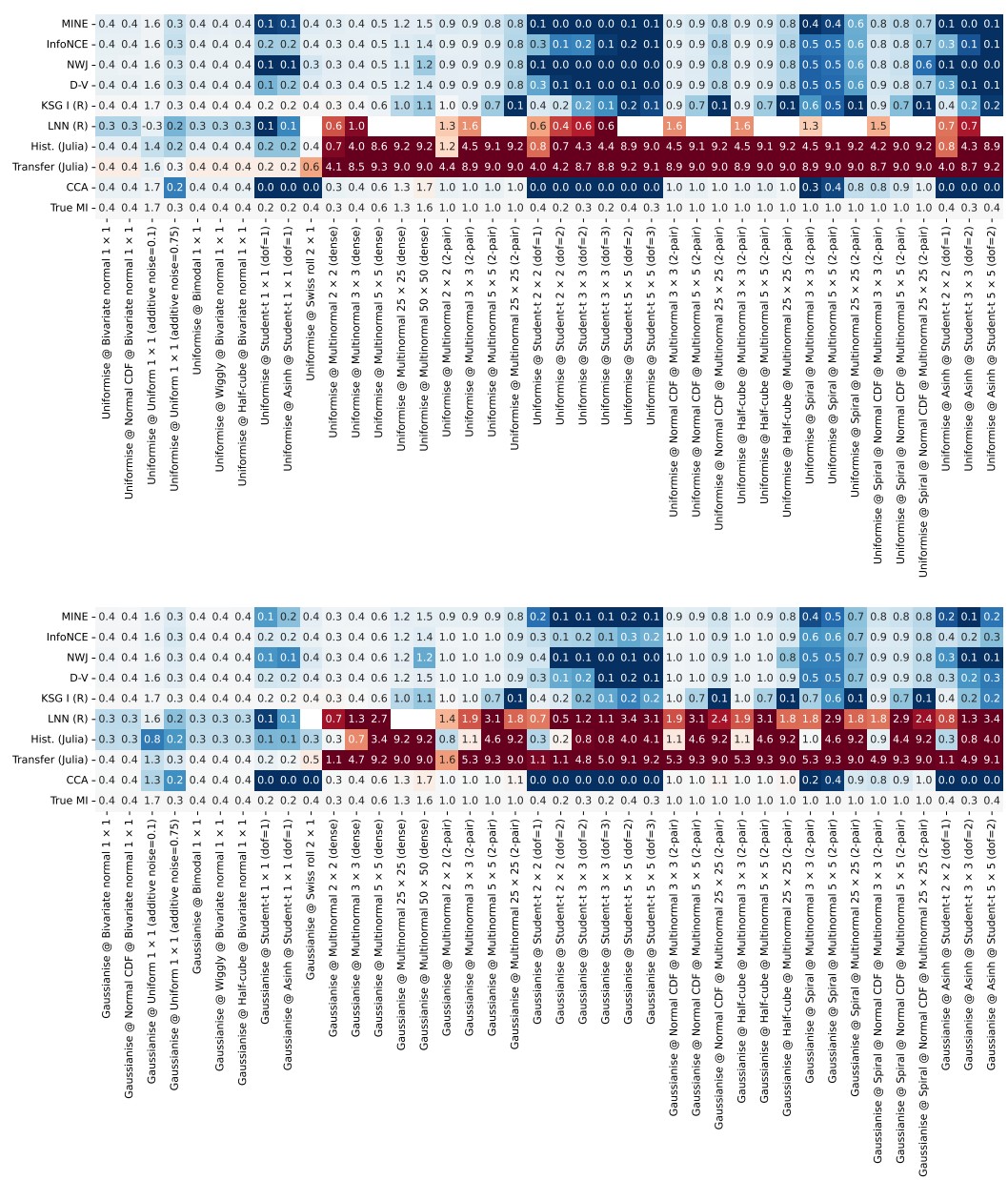

Figure 10: Estimates for preprocessed data with $N = 10\,000$ samples. Top: margins transformed into uniform distributions. Bottom: margins transformed into normal distributions.

We chose the architecture using a mini-benchmark with four ReLU neural networks considered on two neural estimators (see Fig. 11) and chose a neural network with two hidden layers (network M in Table 2).

As neural estimators rely on stochastic gradient descent training, we implemented heuristic diagnostics used to diagnose potential problems with overfitting and non-convergence. We present the number of runs susceptible to these issues in Fig. 12. We did not include these runs when computing statistics in the main benchmark figures (Fig. 2 and Fig. 9).

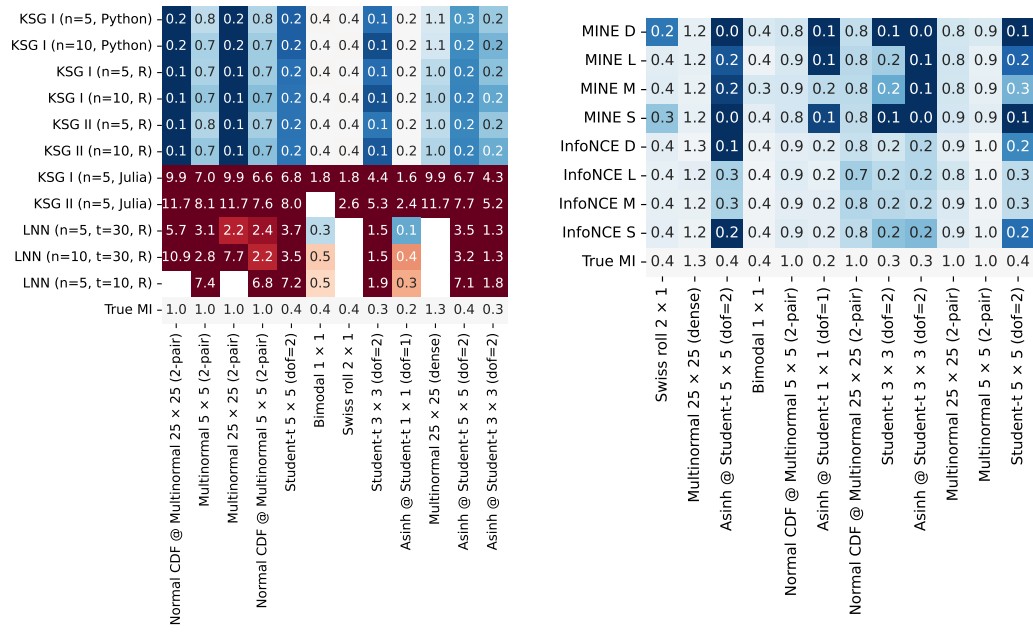

Figure 11: Left: ablation of hyperparameters of the KSG and LNN estimators (with $n$ neighbors and threshold $t$). Right: ablation of neural network architectures.

| Architecture | Hidden layers |
|:---:|:---:|
| D | 8, 8, 8 |
| L | 24, 12 |
| M | 16, 8 |
| S | 10, 5 |

Table 2: Network architectures and their corresponding hidden layer specifications.

## F   Visualisation of Selected Tasks

In the following, we visualise a selection of benchmark tasks. Each figure visualises a sample of $N = 10\,000$ points from $P_{XY}$. The upper diagonal represents the scatter plots $P_{X_i X_j}$, $P_{X_i Y_j}$ and $P_{Y_i Y_j}$ for all possible combinations of $i \neq j$. The lower diagonal represents the empirical density fitted using a kernel density estimator with 5 levels. The diagonal represents the histograms of the marginal distributions $P_{X_i}$ and $P_{Y_i}$ for all possible $i$.

We present selected $1\times1$-dimensional tasks in Fig. 13 and Fig. 14; the $2\times1$-dimensional Swiss-roll distribution in Fig. 15 and selected $2\times2$-dimensional tasks in Fig. 16. Finally, in Fig. 17 we present two $3\times3$-dimensional problems, visualising the effect of the normal CDF and the spiral diffeomorphism.

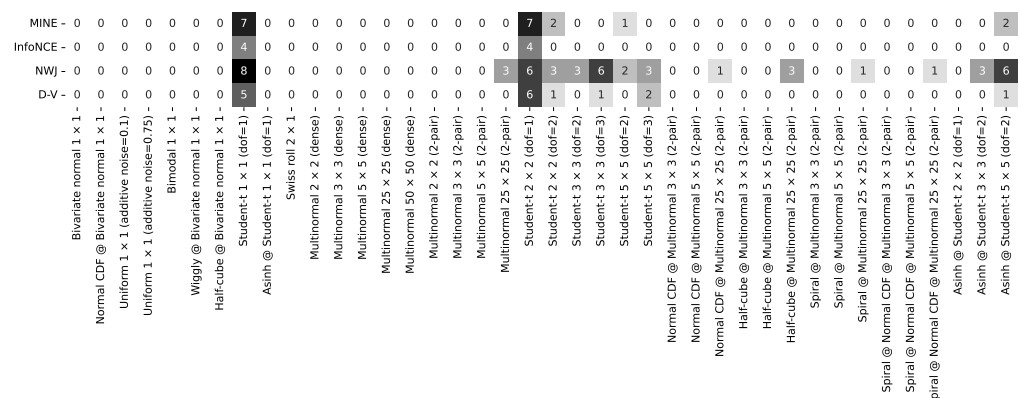

Figure 12: Number of runs (out of 10) when our convergence and overfitting diagnostics diagnosed potential problems in neural estimator training.

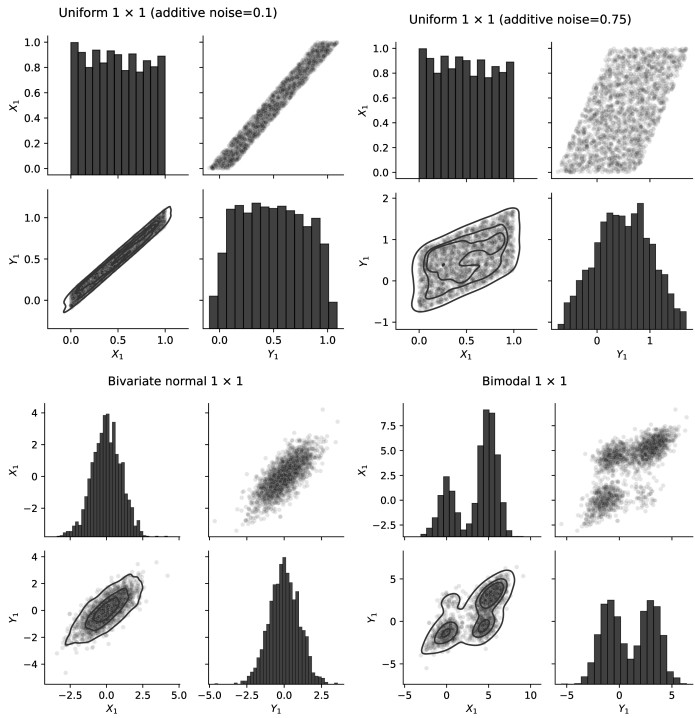

Figure 13: Top row: additive noise model. Bottom row: bivariate normal distribution and transformation making each variable a bimodal mixture of Gaussians.

# G   Author Contributions

Contributions according to the Contributor Roles Taxonomy:

1. Conceptualization: PC, FG, AM, NB.

2. Methodology: FG, PC.

3. Software: PC, FG.

4. Validation: FG, PC.

5. Formal analysis: PC, FG.

6. Investigation: FG, PC, AM.

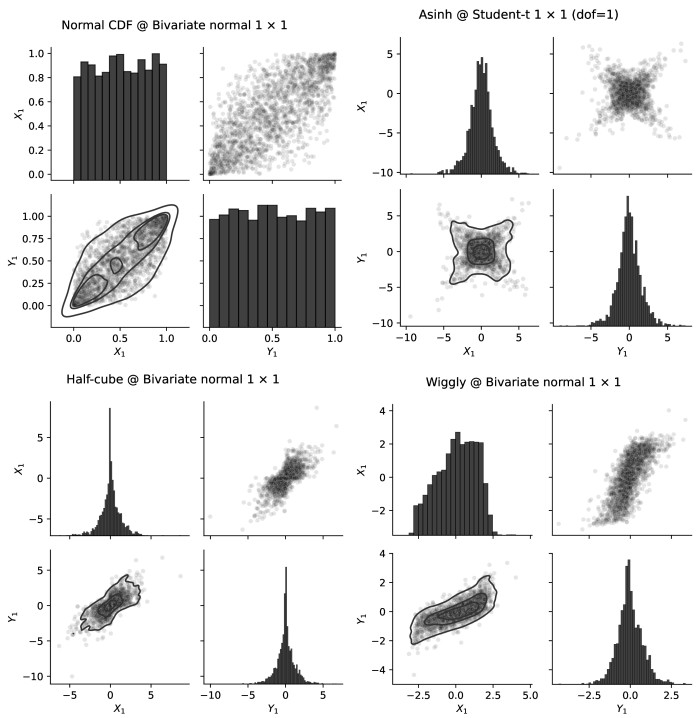

Figure 14: Top row: bivariate normal distribution transformed so that the margins are uniform and asinh-transformed bivariate Student distribution with 1 degree of freedom. Bottom row: bivariate normal distribution transformed with the half-cube and the wiggly mappings.

7. Resources: FG, NB.

8. Data Curation: FG, PC.

9. Visualization: FG, PC.

10. Supervision: AM, NB.

11. Writing – original draft: PC, AM, FG.

12. Writing – review and editing: PC, AM, FG, NB, JEV.

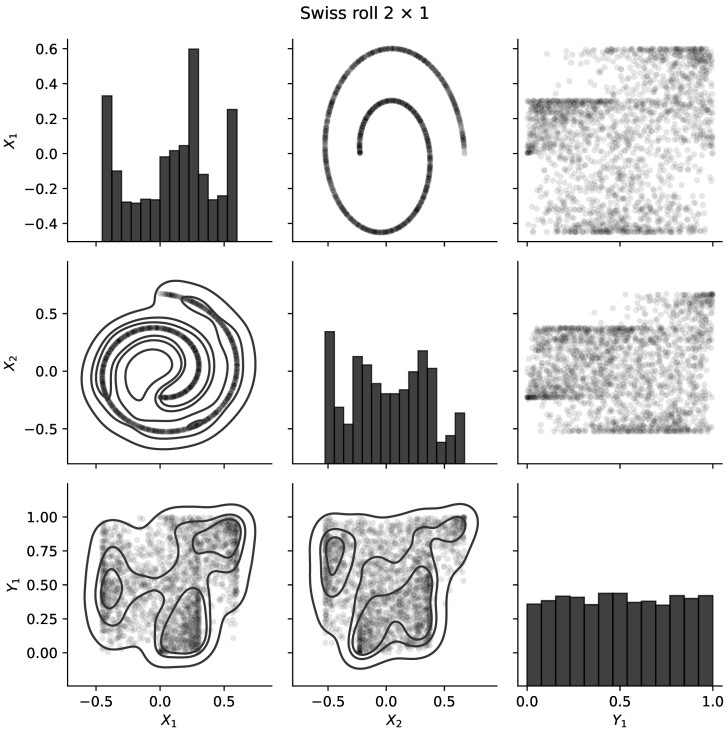

Figure 15: Visualisation of the Swiss-roll distribution.

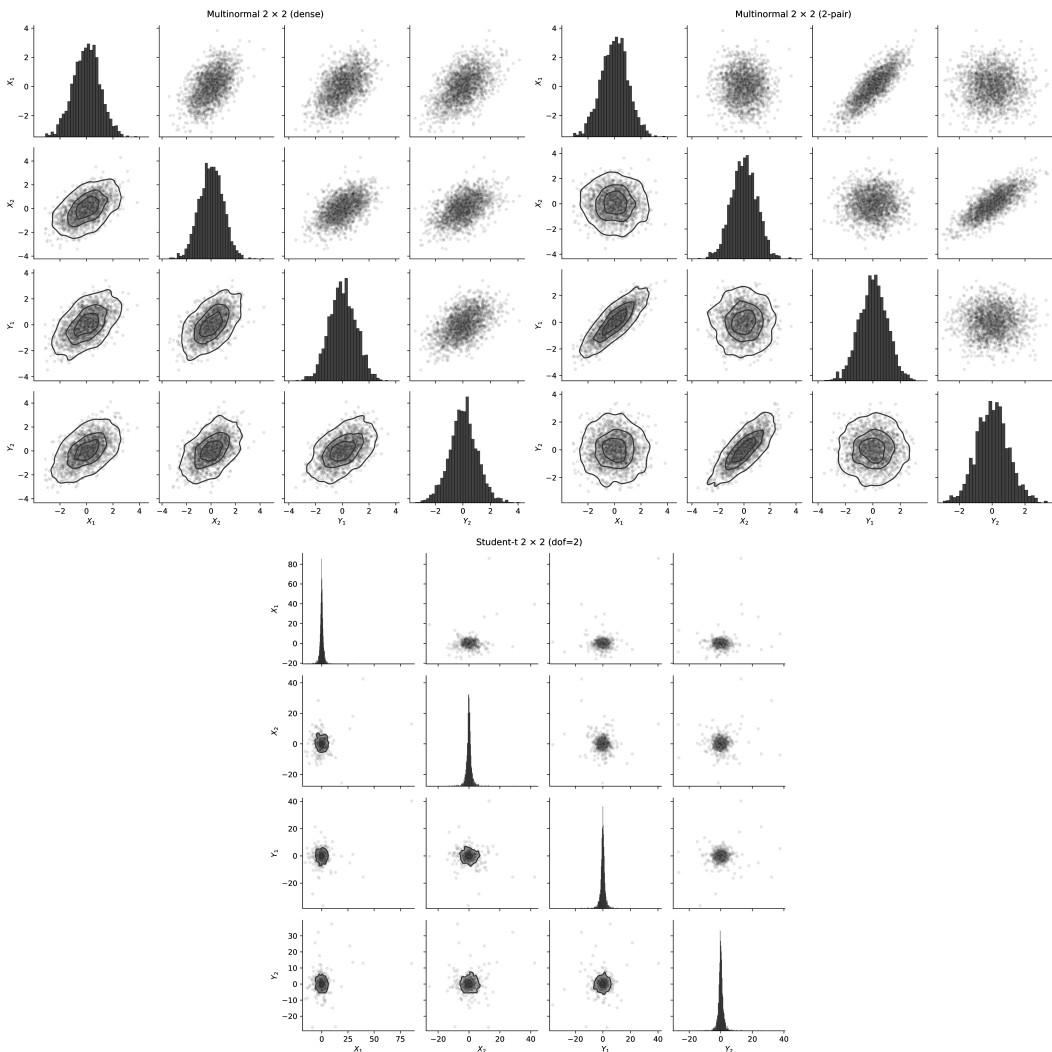

Figure 16: Visualisations of selected $2 \times 2$-dimensional problems. Top row: multivariate normal distributions with dense and sparse interactions. Bottom row: multivariate Student distribution with 2 degrees of freedom.

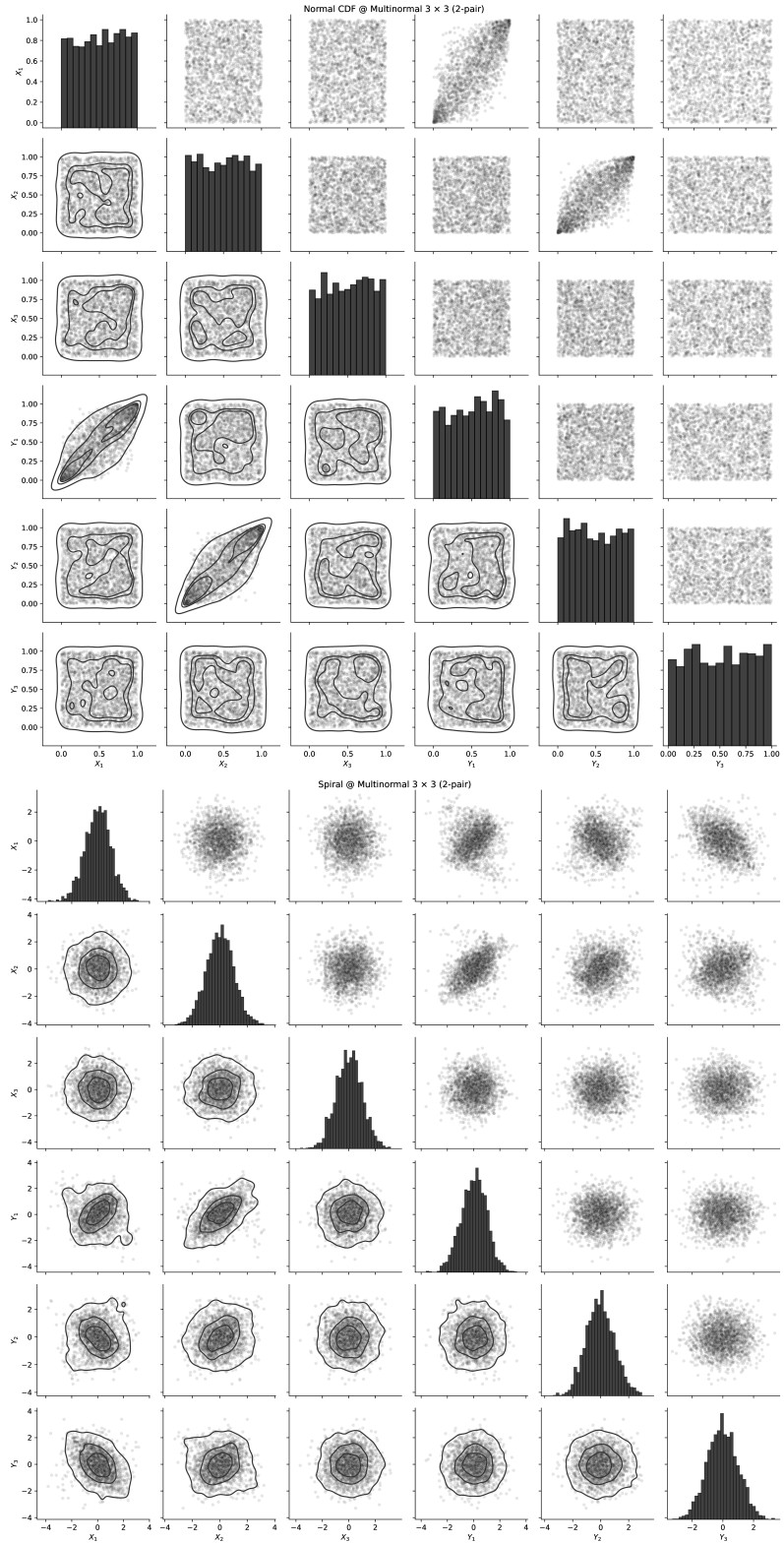

Figure 17: Visualisations of selected $3{\times}3$-dimensional problems. Top: multivariate normal distribution transformed axis-wise with normal CDF, to uniformise margins. Bottom: multivariate normal distribution transformed with the spiral diffeomorphism.

