# OpenReview forum: "Beyond Normal: On the Evaluation of Mutual Information Estimators"
_NeurIPS.cc/2023/Conference — NeurIPS 2023 poster_

### Official Review · Reviewer_62uG · 2023-06-22

**Soundness:** 3 good
**Presentation:** 3 good
**Contribution:** 3 good
**Rating:** 6
**Confidence:** 4

**Summary:**

The authors propose a method for creating expressive distributions via injective mappings that maintain their original MI. They state this in Theorem 2.1 and prove it in the Appendix. In addition to this, the authors benchmark a variety of estimators for MI on a set of tasks, including high-dimensional, long-tailed distributions, sparse interactions, and high mutual information. This information is contained in Sections 3-4, where they describe each task and critique each estimator for the tasks. From these experiments, the authors provide guidelines for practitioners on how to choose an estimator in Section 6.

**Strengths:**

The paper is clear, provides many novel benchmarking tasks, and is easily verifiable. The results are reproducible as the authors have shared their code and documented their experimental parameters.

**Weaknesses:**

1. The author does a poor job of motivating each data setting and explaining why each one is important. It would be beneficial to provide examples of domains where long-tail distributions are common, such as insurance, and elaborate on the significance of the other data settings as well.
2. I believe the last row of Figure 2, "True MI," could be better highlighted as it was difficult to discern that it was the point of comparison.
3. The main contributions of the paper seem relatively minor, as the authors are primarily considering more data settings than previous work when comparing MI estimators. However, addressing the first point could help alleviate this concern.
4. The main theoretical result, Theorem 2.1, appears to have already been demonstrated in the appendix of the following paper: https://arxiv.org/pdf/cond-mat/0305641.pdf.
5. The authors seem to have switched \citep for \cite in their paper.

**Questions:**

Why not utilize the result of Theorem 2.1 and apply it to Normalizing Flows? I believe this combination would enhance the paper's distinctiveness.

How is Theorem 2.1 different from that of the result in the appendix of https://arxiv.org/pdf/cond-mat/0305641.pdf?

**Limitations:**

The authors have addressed the limitations of their work.

---

> ### Author Rebuttal · Authors · 2023-08-08
>
> Thank you for your review.
>
> > The main theoretical result, Theorem 2.1, appears to have already been demonstrated in the appendix of the following paper: Kraskov et al. (2004)  [...] How is Theorem 2.1 different from that of the result in the appendix of Kraskov et al. (2004)?
>
> We believe that Theorem 2.1 has been known in the community and we do not list it as a contribution. We use it to define the distributions used in our benchmark, which is the main content of the paper. However, we could not find an appropriate version of this theorem with a formal proof which we could reference, hence for completeness, we provide our own proof in the Appendix.
>
> The highly influential paper of Kraskov et al. (2004) covers the case where (a) applied mappings are diffeomorphisms and (b) all measures involved have probability density functions. The proof we give in the Appendix applies more generally (and this is needed e.g., for the "Swiss roll" distribution which uses topological embeddings or for the half-cube mapping which is not a diffeomorphism at $0$).
>
> > The author does a poor job of motivating each data setting and explaining why each one is important. It would be beneficial to provide examples of domains where long-tail distributions are common, such as insurance, and elaborate on the significance of the other data settings as well.
>
> The main motivation for selecting the distributions was to evaluate and compare existing estimators. We would like to stress that the space of all possible distributions is vast, and until now, was left almost entirely unexplored except for the simplest case of a normal distribution. This is particularly striking for mutual information, which is commonly used precisely for its applicability to any distribution and invariance to diffeomorphisms.
>
> We approach this issue by casting a wide net of tasks, paying particular attention to the following:
>
> 1. **Dimensionality**. High-dimensional datasets are becoming more common, particularly in machine learning and systems biology.
> 2. **Sparsity**. While the data might be high-dimensional, the effective dimension may be much smaller (e.g., only a few genes out of thousands convey information about the amount of a particular protein).
> 5. **High MI**. Estimating high MI is known to be difficult. However, it is usually something we might know approximately in advance -- if there are 4000 image classes, MI between image class and representation is at most 12 bits. An additional interesting observation is that CCA performes very well, suggesting that in this scenario incorporating prior information is crucial.
> 6. **Long tails**. Since the Student distribution has heavier tails than the multivariate normal distribution, this was a natural choice. An interesting conclusion is that even after "removing the tails" (see Appendix E), these distributions remain difficult. Thus, we have low- and moderate-dimensional distributions, which are unimodal and without heavy tails, which are still challenging to estimate.
> 7. **Robustness to diffeomorphisms**. Mutual information is often chosen because it is theoretically invariant to diffeomorphisms. We wanted to challenge this invariance when only a finite sample is available.
>
> However, we agree that including real-world examples illustrating these motivations will improve the manuscript.
>
> > The main contributions of the paper seem relatively minor, as the authors are primarily considering more data settings than previous work when comparing MI estimators. However, addressing the first point could help alleviate this concern.
>
> This is the first benchmark which can be used to compare different mutual information estimators in a systematic and reproducible manner. As mentioned above, it is critical to evaluate mutual information estimators on non-normal distributions. Our benchmark contains a diverse set of 40 distributions, addressing problems like sparsity of interactions, long tails and invariances to selected mappings.
>
> As such, we believe this manuscript will be useful for the community for two reasons:
>   - For the mutual information researchers, it provides a standard benchmark which can be used to conveniently test their ideas and understand their strengths and limitations. To increase the chances that it is adopted by the community, we made it cross-platform, used well-defined interfaces in the API, and implemented a diverse set of baseline estimators.
>   - For researchers willing to use mutual information in selected problems ranging from machine learning to natural sciences (e.g., Uda (2020), Young et al. (2023)), a set of points to be aware of when a reliable estimate is needed.
>
> > I believe the last row of Figure 2, “True MI,” could be better highlighted as it was difficult to discern that it was the point of comparison. [...] The authors seem to have switched \citep for \cite in their paper.
>
> Thank you for these suggestions: we have improved the figure accordingly and fixed inconsistency with citations.
>
> > Why not utilize the result of Theorem 2.1 and apply it to Normalizing Flows? I believe this combination would enhance the paper’s distinctiveness.
>
> Indeed, using normalizing flows is an interesting idea which can be used as:
>   1. Transformations to define more expressive distributions.
>   2. Another preprocessing technique (similar to the ones studied in Appendix E).
>
> We had considered both options, but eventually, we have decided to use interpretable alternatives: proposed mappings were able to capture several interesting phenomena already, and we were concerned that using normalizing flows may distort the distributions in unexpected manners, making it difficult to separate failure modes of different estimators.
>
> We hope that our response provided an additional perspective on the significance of this work and that the introduced changes have increased the manuscript's clarity. Hence, we would like to kindly ask the Reviewer to consider raising the scores.

---

> > ### Comment · Reviewer_62uG · 2023-08-16
> >
> > Thank you for your response.
> >
> > Theorem 2.1:
> >
> > I appreciate your effort in providing further clarification. Nevertheless, I maintain my belief that the novelty of the theorem might still appear ambiguous to readers. To enhance its clarity, I suggest considering referencing the proof by Kraskov et al. (2004) or another similar result. Despite its familiarity within the community, it's important to remember that individuals from outside the community might perceive this as novel.
> >
> > Motivating data settings:
> >
> > I commend the authors for their thorough approach in encompassing a diverse range of distributions. Thus making their benchmark applicable to a large set of domains..
> >
> > My primary reason for the initial low score pertains to the treatment of Theorem 2.1. If the authors could furnish additional context surrounding Theorem 2.1, I would happily raise my evaluation. I appreciate the authors for their considerate response and answers to all my questions.

---

> > > ### Author Response · Authors · 2023-08-17
> > >
> > > Thank you for your suggestion and encouraging words on the diverse range of distributions covered!
> > >
> > > We have now understood your argument regarding Theorem 2.1 and we fully agree that adding more context to it will significantly increase manuscript clarity. We will add the following paragraph to Section 2:
> > >
> > > > Theorem 2.1 is a well-known property of mutual information, formulated in various versions. For example, Kraskov et al. (2004) consider a case in which $f$ and $g$ are diffeomorphisms and all measures have probability density functions. For the sake of completeness, we include a proof of Theorem 2.1 (covering singular measures and any continuous injections) in Appendix A.
> > >
> > > Please, let us know if you have any further suggestions.

---

> > > > ### Comment · Reviewer_62uG · 2023-08-17
> > > >
> > > > Thank you for the change, I have changed my score to weak accept. Good luck!

---

### Official Review · Reviewer_vCqe · 2023-06-23

**Soundness:** 4 excellent
**Presentation:** 4 excellent
**Contribution:** 3 good
**Rating:** 7
**Confidence:** 3

**Summary:**

This paper identifies a clear problem with many mutual information estimation benchmarks: most of them focus on simple (normal) distributions. The authors present a new set of forty (!) tasks that contain ground-truth informations, which can be constructed by noting that only injectivity is needed for an information-preserving transformation. The authors identify four distinct challenges: interaction sparsity, long tails, invariance, and high information values. Several conclusions about existing estimation algorithms are then made regarding the extensive analysis.

**Strengths:**

- I enjoyed reading this paper. It clearly defines a goal and identifies key problems with existing approaches.
- The paper presents mathematical background in a precise and effective way.
- The structure of the paper is clear, using figures for clarifications where needed.
- The paper is well-written with clear sentence structures and no grammatical or spelling errors.
- The analysis is thorough, presenting forty distinct MI estimation tasks.
- I enjoyed how instead of simply presenting the results, the authors dug deeper and identified four distinct challenges for MI estimators.


**Weaknesses:**

- One could say that the paper lacks slightly in terms of originality and contributions.

**Questions:**

- Out of curiosity: could modern invariant neural network architectures be used to obtain MI estimates invariant to diffeomorphisms?


### Conclusions
While the overall contribution could be limited in terms of a model development sense, I think the paper identifies serious issues with modern MI estimation benchmarks. The paper not only provides new benchmarks that address these issues but also makes an effort to identify what aspects of MI estimation can make the problem hard. I foresee much new research originating from the identification of these aspects, where future papers focus in on them and propose methodologies that overcome these challenges. On top of that, the paper is very well written. Hence, I would recommend acceptance.

**Limitations:**

The authors clearly discuss the limitations of their study.

---

> ### Author Rebuttal · Authors · 2023-08-08
>
> Thank you for the positive assessment and encouraging words.
>
> > Out of curiosity: could modern invariant neural network architectures be used to obtain MI estimates invariant to diffeomorphisms?
>
> Thank you for your insightful question! Sadly, we cannot achieve full invariance to diffeomorphisms for finite samples since an arbitrary diffeomorphism can transform any set of $n$ points to any other set of $n$ points in $\mathbb R^k$ when $k\ge 2$. Nonetheless, we can hope for invariance (or robustness) to a subset of diffeomorphisms. We hypothesise that such invariant neural networks could require smaller sample sizes due to their useful inductive biases. However, we have not experimented with invariant neural networks ourselves, so we are unable to support this hypothesis with data.
>
> We think that there are several solutions how this problem can be approached: either by encoding invariance to, e.g., the group of rigid motions by appropriate layers (in this case based on the representation theory of the orthogonal group) or using training schemes increasing robustness to diffeomorphisms, akin to [1, 2]. Our benchmark can be used to generate data sets for the latter approach, however thorough investigation of this idea is beyond the scope of this work.
>
> [1] Benton, Gregory, et al. "Learning invariances in neural networks from training data." Advances in neural information processing systems 33 (2020): 17605-17616.
>
> [2] Petrini, Leonardo, et al. "Relative stability toward diffeomorphisms indicates performance in deep nets." Advances in Neural Information Processing Systems 34 (2021): 8727-8739.

---

> > ### Comment · Reviewer_vCqe · 2023-08-16
> >
> > Thank you for providing the detailed response. Your elaboration on the potential applications and limitations of invariant neural networks in relation to diffeomorphisms is enlightening. Thanks for the references.

---

### Official Review · Reviewer_FjRx · 2023-06-30

**Soundness:** 4 excellent
**Presentation:** 4 excellent
**Contribution:** 2 fair
**Rating:** 5
**Confidence:** 4

**Summary:**

A test benchmark for the evaluation of mutual information estimators is established and many different estimators compared. The test cases contain student-t and normal distributionas and their injective transformations. Difficult cases are discussed and evaluated in more detail.

**Strengths:**

The code is reproducible and thus might be used for other estimators in the future
The paper is well-written and easy to understand.
It is important to make the community aware that evaluation of MI estimators on Gaussian distributions is pointless as these only depend on the covariance structure, so the big plus of the MI that it goes beyond te linear dependencies is ignored. The paper makes a strong point here by including a simple covariance estimator as well.
The results on heavy tails are particularly interesting.

**Weaknesses:**

The choice of the distribution used is not sufficiently argued. In particular, it is known that no MI estimator can evaluate MI correctly on arbitrary distributions. Only with restriction to a class of probability distributions (e.g., probability density functions with Lipschitz constraints) there is hope that estimation works. It is thus quite pointless to evaluate MI estimators on what seems like an "educated guess" of diverse distributions.
The authors seem to not be aware of the huge theoretic background of MI, e.g., that arbitrary measureable injective mappings do not change MI and thus their only Theorem 2.1 is well-known in a much more general setting. This can be argued by the data processing inequality in two directions (X-Y-f(Y) and X-f(Y)-Y are both Markov chains) or directly the definition of MI via countable partitions that do not change if we use a measureable injective mapping.


**Questions:**

What was the reason for choosing this specific set of distributions?
What is "pointwise MI"?

**Limitations:**

The authors state that there are more interesting cases and they only cover transforms of normal and student-t distributions. They also mention that prior information might be incorporated.
However, as said above it is known that an MI estimator can be fooled arbitrarily (giving any value for the MI) if one can choose the distribution freely. Thus, prior information is also included in the test cases here it is just not mentioned explicitly.

---

> ### Author Rebuttal · Authors · 2023-08-08
>
> Thank you for the detailed comments.
>
> > The choice of the distribution used is not sufficiently argued. In particular, it is known that no MI estimator can evaluate MI correctly on arbitrary distributions. (...)
>
> Indeed, one version of a no-free-lunch theorem for MI estimation follows from the fact that for $n\ge 1$ and $k\ge 2$ any set of $n$ points in $\mathbb R^k$ can be mapped to any other set of $n$ points by a diffeomorphism (if $M$ is any connected smooth manifold of dimension at least $2$, the group $\mathrm{Diff}(M)$ acts $n$-transitively on it). It is thus clear that the only truly diffeomorphism-invariant estimators have to be constant functions (for a given number of points).
> Nonetheless, invariance to diffeomorphisms is precisely the reason why MI is useful in practice. The theorem above implies that all estimators have to break at some point. However, while no estimator can work on all distributions, different estimators can work better on certain types of distributions. From a practical standpoint, it is interesting whether certain distributions which intuitively seem "reasonable" are difficult for estimators (e.g., even low-dimensional Student distributions are challenging for most estimators), as well as which estimators are particularly suitable for which types of problems (e.g., neural estimators are good at solving problems involving sparse interactions).
>
> The issue of which distributions are "reasonable" is indeed vague. Certainly, some distributions are "unreasonable" (e.g., a complicated embedding into a 1000-dimensional space), and we do not expect any estimator to solve them. In our study, we avoid such uninformative problems by excluding tasks which no estimator was able to solve. We also include visualisations of the considered distributions to allow for visual inspection (Appendix F).
>
> > What was the reason for choosing this specific set of distributions?
>
> As explained above, the goal was to understand the limits and advantages of different estimators. To this end, we decided to focus on relevant phenomena (e.g., robustness to "sufficiently nice" diffeomorphisms, sparsity) by designing a set of transformations wide enough to (a) understand the mentioned issues and (b) construct a standardised benchmark, which can be used to diagnose strengths and weaknesses of new estimators.
>
> For the motivations of individual phenomena, see the General Response.
>
> > They also mention that prior information might be incorporated. However, as said above it is known that an MI estimator can be fooled arbitrarily (giving any value for the MI) if one can choose the distribution freely. Thus, prior information is also included in the test cases here it is just not mentioned explicitly.
>
> Yes, since no estimator can work on all distributions, each estimator can be thought of as having an implicit bias towards distributions it can handle well. For example, we show that the KSG estimator is not competitive with neural approaches when interactions are sparse. Thus, if we believe that the distribution we are analyzing has sparse interactions, we should use a neural approach. We argue that using (and developing) estimators with explicitly known assumptions and biases could result in significant performance gains. We demonstrate this by implementing CCA (which uses very strong prior information), and showing that it is an excellent choice when the distribution matches assumptions.
>
> We think the term "prior information" in the manuscript may have been misleading and not convey the above meaning. We will clarify it in the manuscript.
>
> > The authors seem to not be aware of the huge theoretic background of MI, e.g., that arbitrary measureable injective mappings do not change MI and thus their only Theorem 2.1 is well-known in a much more general setting. This can be argued by the data processing inequality in two directions (X-Y-f(Y) and X-f(Y)-Y are both Markov chains) or directly the definition of MI via countable partitions that do not change if we use a measureable injective mapping.
>
> We believe that Theorem 2.1 has been known in the community for a long time and therefore we did not list it as a contribution. In spite of our efforts, we were not able to find a reference with a formal proof of the theorem which covers singular measures, infinite MI and topological embeddings. Since we use this result extensively, we supplied an appropriate version in the appendix. We would be grateful for suggesting a reference covering this (or a more general) result, so that we can cite it. The proof provided by Kraskov et al. (2004), for example, does not cover the “Swiss roll” distribution, which uses a topological embedding or the half-cube mapping, which is not a diffeomorphism at 0.
>
> > What is pointwise MI?
>
> We apologise for using the term "pointwise MI" in Section 6 rather than "pointwise mutual information", which we use in all the other sections. We fixed this notational inconsistency.
>
> > The authors state that there are more interesting cases and they only cover transforms of normal and student-t distributions.
>
> There are many distributions which cannot be obtained as $P_{f(X)g(Y)}$, where the join $(X, Y)$ is a multivariate normal or Student distribution and $f$, $g$ are continuous injective mappings. However (except for some simple bivariate distributions and trivial cases with zero MI), an analytical expression for non-zero ground-truth MI is currently tractable only for the multivariate normal and Student families.
>
> We consider enlarging the proposed family an important open direction of the problem and we designed our benchmark so that adding new distributions with known ground-truth MI requires little effort.
>
> We hope that we were able to answer questions raised in the review and clarify our contributions. Given that the Reviewer confirmed excellent soundness and presentation, and a fair contribution, we would be thankful if the Reviewer could reconsider the overall score.

---

> > ### Comment · Reviewer_FjRx · 2023-08-11
> >
> > Thank you for the detailed response.
> >
> > Indeed, I had a hard time finding a source for Theorem 2.1 as well and can only provide data processing inequalities that then imply the authors result as corollary. I also have to admit that mere measurability as conjectured in my original review is not sufficient and some way to show measureability of the inverse (defined on the range) is also required.
> >
> > I'm still reluctant to support the given set of distributions as reasonably representative but have to admit that so far MI has been estimated on much worse datasets, thus it is a step in the right direction. I also question the statement that "an analytical expression for non-zero ground-truth MI is currently tractable only for the multivariate normal and Student families." With some basic math, other approches like sums of uniform distributions should also be tractable.
> >
> > Nevertheless, I'll increase my score as the paper is a step in the right direction but hope that soon more versatile distributions will be added to the benchmark and we are not stuck with a less than perfect solution for the next decade.

---

> > > ### Author Response · Authors · 2023-08-14
> > >
> > > Thank you very much for your response. We agree that the chosen set of distributions is not universal, but, at the same time, is a step in the right direction.
> > > Regarding the sums of uniform distributions, we would like to thank you for the suggestion. We have already included in our benchmark the bivariate case (lines 134–137): $Y = X+N$, where $X\sim \mathrm{Uniform}(0, 1)$ and $N\sim \mathrm{Uniform}(-\varepsilon, \varepsilon)$, but we agree that a multivariate generalization (with independent $X_1, \dotsc, X_k$ and $N_1, \dotsc, N_k$) has tractable ground-truth mutual information as well. We will add it to the benchmark.

---

### Official Review · Reviewer_AkdF · 2023-07-07

**Soundness:** 3 good
**Presentation:** 3 good
**Contribution:** 3 good
**Rating:** 6
**Confidence:** 3

**Summary:**

This paper focuses on the topic of mutual information and shows how to construct a diverse family of distributions with known ground-truth mutual information. It's worth noting that obtaining a closed-form solution for mutual information is highly dependent on the specific assumptions and functional forms used for the variables X and Y. In practice, deriving closed-form expressions for mutual information can be challenging and may require additional simplifying assumptions or specific knowledge about the distributions involved.

In contrast to previous works that typically assess mutual information estimators using simple probability distributions, this paper introduces a novel approach to constructing a diverse family of distributions with known ground-truth mutual information. Additionally, the authors propose a language-independent benchmarking platform to assess mutual information estimators. The authors explore the applicability of classical and neural estimators in scenarios involving high dimensions, sparse interactions, long-tailed distributions, and high mutual information. By examining these challenging settings, they provide insights into the strengths and limitations of different estimators.

Moreover, the paper offers guidelines for practitioners to select the most suitable estimator based on the specific problem's difficulty and considerations when applying an estimator to new datasets. By presenting a comprehensive evaluation framework and practical recommendations, this research aims to advance the understanding and application of mutual information estimation in various domains.

**Strengths:**

The mutual information estimator is an essential tool in causality, it can help discover the underlying causal graph or inference the strength of causal relations. However, as I mentioned earlier, deriving closed-form expressions for mutual information can be challenging in practice and may require additional simplifying assumptions or specific knowledge about the distributions involved.

The paper introduces a method to construct a diverse family of distributions with known ground-truth mutual information. This is a significant contribution as it allows researchers to explore and evaluate mutual information estimators across various scenarios, encompassing various data characteristics and relationships. For example, explore gene regularity networks, understand the treatment effect in medical health care, gain insight for constructing a recommendation system, etc.

The research paper presents a comprehensive evaluation framework for mutual information estimation, encompassing the construction of diverse distributions, benchmarking platform, exploration of challenging scenarios, and practical guidelines. This framework provides a holistic view of the estimation process, aiding researchers and practitioners in understanding, comparing, and selecting mutual information estimators effectively.

Furthermore, the authors investigate the applicability of classical and neural estimators in challenging scenarios involving high dimensions, sparse interactions, long-tailed distributions, and high mutual information. This exploration provides valuable insights into the performance, strengths, and limitations of different estimators under these challenging conditions, enhancing our understanding of their effectiveness in real-world settings.

**Weaknesses:**

1. Not all joint distributions can be represented in the form of $P_{f(x)g(x)}$, limiting the applicability of the benchmark to a specific set of distributions. Extending the family of distributions with known mutual information and efficient sampling is seen as a natural direction for future improvement.
2. Even though the benchmark demonstrates that distributions with longer tails pose a harder challenge for the considered estimators, applying a transformation like the asinh transform does not fully address the issues.
3. The summary does not mention external validation or comparisons between the proposed approach or estimators and existing methods or benchmarks in the field. The absence of such external validation makes it difficult to assess the generalizability or superiority of the contributions in relation to established techniques or alternative approaches.

**Questions:**

See above "weaknesses".

**Limitations:**

See above "weaknesses".

---

> ### Author Rebuttal · Authors · 2023-08-08
>
> Thank you for your thorough review. Regarding the questions and limitations:
>
> > Not all joint distributions can be represented in the form of $P_{f(X)g(Y)}$, limiting the applicability of the benchmark to a specific set of distributions. Extending the family of distributions with known mutual information and efficient sampling is seen as a natural direction for future improvement.
>
> Yes, we agree with this assessment. As noted by the Reviewer, obtaining a closed-form solution for mutual information usually requires simplifying assumptions, which constrain the distributions.
>
> One way to extend the family of distributions is to allow for random variables for which mutual information can be efficiently estimated numerically. For example, when the joint and marginal probability distributions have tractable PDFs, mutual information can be estimated by averaging pointwise mutual information over a sufficient amount of Monte Carlo samples. In most cases, the standard error of the mean should be a sufficient diagnostic to provide precise and accurate estimate. This type of task could be readily implemented in our package.
>
> >  Even though the benchmark demonstrates that distributions with longer tails pose a harder challenge for the considered estimators, applying a transformation like the asinh transform does not fully address the issues.
>
> Indeed, our experiments suggest that although long tails make estimation harder, simple approaches of "shrinking" the tails (as with the asinh transform or different preprocessing strategies explored in Appendix E) cannot resolve this issue completely. We consider this an interesting open problem and suspect that this may be related to the shape of the "PMI profile" (defined below).
>
> Let $i$ be the pointwise mutual information (PMI) function for distribution $P_{XY}$. We define the PMI profile to be the distribution of $i(X, Y)$. This distribution is defined on the set of real numbers and its expected value is the mutual information. Higher moments (and, informally, the overall "shape") of this distribution can influence how well the expected value can be estimated. Importantly, the PMI profile does not change under diffeomorphisms, so the PMI profiles of $P_{XY}$ and $P_{f(X)g(Y)}$ are the same.
>
> We hypothesise that this may explain why these simple preprocessing strategies (which proceed by applying diffeomorphisms) cannot fully resolve the issues with long-tailed distributions, but we have not validated this hypothesis yet.
>
> > The summary does not mention external validation or comparisons between the proposed approach or estimators and existing methods or benchmarks in the field. The absence of such external validation makes it difficult to assess the generalizability or superiority of the contributions in relation to established techniques or alternative approaches.
>
> A lack of universally accepted, easy-to-use benchmarks for testing mutual information estimators is one of our main motivations. One of the packages implementing several estimators we consider in our benchmark was covered by unit-tests which only ensured that the returned value is a `float`(!). Other benchmarks (Song and Ermon (2020), Khan et al. (2007), and Poole et al. (2019), which are discussed in Section 5) are typically constructed for evaluating a specific class of estimators, and usually focus on Gaussian variables (with known MI) or complicated high-dimensional tasks (with MI not known). This has made it difficult to build a general understanding of available methods and their applicability. We believe that our benchmark can serve as a strong reference point for future work.

---

### Author Rebuttal · Authors · 2023-08-08

We would like to thank the Reviewers for their insightful comments and appreciate that they find that our work *"provides valuable insights into the performance, strengths, and limitations of different estimators*" (AkdF), that our *"results on heavy tails are particularly interesting"* and that the paper *"makes a strong point here by including a simple covariance estimator as well"* (FjRx). It *"presents mathematical background in a precise and effective way"* (vCqe) and that *"the paper is clear, provides many novel benchmarking tasks, and is easily verifiable"* (62uG). We find it particularly encouraging that all reviewers find that our paper is clear and that Reviewer vCqe concludes *"I foresee much new research originating from the identification of these aspects"*.

Further, we would like to comment on two points asked by both Reviewer FjRx and Reviewer 62uG.

Regarding the motivation of used distributions, we approach the issue of benchmark construction by casting a wide net of tasks. We decided to focus on the following phenomena:

1. **Dimensionality**. High-dimensional datasets are becoming more common, particularly in machine learning and systems biology.
2. **Sparsity**. While the data might be high-dimensional, the effective dimension may be much smaller (e.g., only a few genes out of thousands convey information about the amount of a particular protein).
5. **High MI**. Estimating high MI is known to be difficult. However, it is usually something we might know approximately in advance -- if there are 4000 image classes, MI between image class and representation is at most 12 bits. An additional interesting observation is that CCA performes very well, suggesting that in this scenario incorporating prior information is crucial.
6. **Long tails**. Since the Student distribution has heavier tails than the multivariate normal distribution, this was a natural choice. An interesting conclusion is that even after "removing the tails" (see Appendix E), these distributions remain difficult. Thus, we have low- and moderate-dimensional distributions, which are unimodal and without heavy tails, which are still challenging to estimate.
7. **Robustness to diffeomorphisms**. Mutual information is often chosen because it is theoretically invariant to diffeomorphisms. We wanted to challenge this invariance when only a finite sample is available.


Secondly, we would like to note that our main contribution lies in constructing a general benchmark and the analysis of its results; we do not consider Theorem 2.1 (a necessary tool to construct our benchmark) to be a novel contribution of this paper (Reviewers FjRx, 62uG), as we feel it has been known by the community for a long time. However, we could not find the proof in the literature (and the proof given by Kraskov et al. (2004), e.g., does not cover singular measures and mappings other than diffeomorphisms), so we included our proof in the Appendix for completeness.

---

### Decision · Program_Chairs · 2023-09-21

**Decision:**

Accept (poster)

**Comment:**

This paper considers the important problem of estimating the mutual information (MI) between two variables. The main contribution is a benchmark for MI estimators with known ground-truth that encompasses several relevant phenomena: dimensionality, sparsity, and heavy tails. This benchmark could have substantial long-term impact as more sophisticated estimators are constructed in the future.